# Network structure of brain atrophy in de novo Parkinson's disease

Yashar Zeighami[1], Miguel Ulla[1,2], Yasser Iturria-Medina[1], Mahsa Dadar[1], Yu Zhang[1], Kevin Michel-Herve Larcher[1], Vladimir Fonov[1], Alan C Evans[1], D Louis Collins[1], Alain Dagher[1]*

[1]McConnell Brain Imaging Centre, Montreal Neurological Institute, McGill University, Montreal, Canada; [2]Service de Neurologie A, CHU Clermont-Ferrand, Clermont-Ferrand, France

**Abstract** We mapped the distribution of atrophy in Parkinson's disease (PD) using magnetic resonance imaging (MRI) and clinical data from 232 PD patients and 117 controls from the Parkinson's Progression Markers Initiative. Deformation-based morphometry and independent component analysis identified PD-specific atrophy in the midbrain, basal ganglia, basal forebrain, medial temporal lobe, and discrete cortical regions. The degree of atrophy reflected clinical measures of disease severity. The spatial pattern of atrophy demonstrated overlap with intrinsic networks present in healthy brain, as derived from functional MRI. Moreover, the degree of atrophy in each brain region reflected its functional and anatomical proximity to a presumed disease epicenter in the substantia nigra, compatible with a trans-neuronal spread of the disease. These results support a network-spread mechanism in PD. Finally, the atrophy pattern in PD was also seen in healthy aging, where it also correlated with the loss of striatal dopaminergic innervation.

## Introduction

Parkinson's disease (PD) is the second most common neurodegenerative illness. Its clinical hallmarks are due to loss of dopamine neurons in the substantia nigra (SN); however, post-mortem studies have shown that PD pathology affects several other brain areas (*Goedert et al., 2013*). Surprisingly, however, magnetic resonance imaging (MRI) studies have failed to consistently demonstrate regional brain atrophy at least in the earlier stages of the disease. This is possibly due to studies consisting mostly of small numbers of subjects from single centers and the use of methodological tools that are relatively insensitive to subcortical atrophy, coupled with univariate methods that suffer from lack of statistical power.

The distribution of Lewy body pathology in post-mortem samples led Braak et al. to propose that a disease process spreads from the brainstem to subcortical areas and then to the cerebral hemispheres along neuronal pathways (*Braak et al., 2003*, *2006*; *Goedert et al., 2013*). More generally, the network-spread hypothesis suggests that all neurodegenerative diseases target spatially distributed intrinsic brain networks (*Warren et al., 2013*). At a molecular level, neurodegenerative diseases are now thought to involve prion-like spread of toxic misfolded protein aggregates (*Frost and Diamond, 2010*; *Jucker and Walker, 2013*). Alpha-synuclein fibrils, implicated in PD pathogenesis, have recently been shown to spread from cell to cell via neuronal pathways after inoculation in mouse brain (*Luk et al., 2012*; *Masuda-Suzukake et al., 2013*).

Neuroimaging studies have revealed that the spatiotemporal organization of the brain takes the form of intrinsic connectivity networks (ICNs) (*Fox and Raichle, 2007*; *Smith et al., 2009*), which are consistent across studies, in test–retest analysis, and during both rest and task states (*Damoiseaux et al., 2006*; *Smith et al., 2009*; *Zuo et al., 2010*). Intrinsic networks have also been thought to be potential targets

*For correspondence: alain.dagher@mcgill.ca

Competing interests: The authors declare that no competing interests exist.

**eLife digest** Although Parkinson's disease is the second most common neurodegenerative disorder, its cause is not known and there is no cure. The symptoms of Parkinson's disease, which include tremor and slowing of voluntary movements, get progressively worse over time. The numbers of neurons in certain brain regions also decrease, causing those parts of the brain to shrink; this is known as 'atrophy'. However, no conclusive signs of atrophy have been found in the brains of people in the early stages of the disease.

One theory suggests that Parkinson's disease is caused by a toxic protein that is able to spread from neuron to neuron. Recent advances in brain imaging have made it possible to map networks in the living human brain—the so-called brain connectome. These networks could form the 'highways' through which a disease-causing agent might spread.

The Parkinson's Progression Markers Initiative (PPMI) is a large study that collects data from hundreds of people in an effort to identify the causes of Parkinson's disease. Zeighami et al. have now analyzed MRI scans that were collected as part of this initiative, which show the structure of the brains of 230 people in the early stages of Parkinson's disease. Comparing these scans to those from age-matched healthy individuals allowed Zeighami et al. to identify the set of brain regions that show atrophy in the early stages of Parkinson's disease. These regions correspond to a normal brain network, and the relative extent of atrophy in each brain region supports the theory that the disease spreads through the connectome.

The patients who were enrolled in this study will continue to be evaluated on a yearly basis. Zeighami et al. plan to continue mapping how the disease progresses throughout the brain and to relate this to the development of new symptoms of Parkinson's disease.

for AD since the earliest reports of default mode network dysfunction in the disease (*Greicius et al., 2004*), and analysis of large MRI data sets in AD and other dementias have supported the network-vulnerability hypothesis (*Seeley et al., 2009*; *Raj et al., 2012*; *Zhou et al., 2012*); however, it has yet to be tested in PD. Here, we analyzed data from newly diagnosed PD patients (n = 232) and an age-matched control group (n = 117) obtained from the Parkinson's Progression Markers Initiative (PPMI) database (www.ppmi-info. org/data), an observational, multicenter longitudinal study designed to identify PD progression biomarkers (*Marek et al., 2011*). We used deformation-based morphometry (DBM) and tensor probabilistic independent component analysis (ICA) to identify brain regions demonstrating atrophy in early PD.

We also sought to provide support for the network-spread hypothesis in PD by showing that the disease, in humans, targets intrinsic brain networks. We compared the set of atrophic regions in PD patients to ICNs from young healthy subjects and tested them for spatial overlap. To further investigate the model of disease propagation through brain networks, we used resting state functional MRI (fMRI) and diffusion-weighted MRI (DW-MRI) of healthy subjects to define the normal brain connectome and determined whether the pattern of atrophy in PD was compatible with spread via this network from a presumed disease epicenter in the SN.

## Results

### ICA detects a PD-specific pattern of deformation

DBM was used as the measure of local brain atrophy. It is a measure of the change in the shape of each voxel that results from applying non-linear spatial normalization of the entire brain to a standard template (*Aubert-Broche et al., 2013*). For each subject, we obtained one parametric image of MRI-determined voxel-wise difference in volume, compared to the template brain. ICA was conducted on these DBM images using FSL MELODIC software (*Beckmann and Smith, 2004*). No constraint was imposed on the number of components, and probabilistic ICA estimated 30 independent components of deformation in the PPMI data set (PD patients and controls combined). Each ICA component consists of a spatial map and the average deformation value in that map for each subject.

In each of these 30 independent components, the average deformation between PD and control groups was compared using an unpaired t-test. PD patients had significantly lower DBM values in one and only one of the 30 deformation components (p = 0.0001 uncorrected, p = 0.003 with Bonferroni

correction). The next spatial component in terms of statistical significance (p = 0.06, uncorrected) consisted of cerebellar areas VIIIA, VIIB, CrusII, and IX known for their involvement in motor and executive function (*Stoodley and Schmahmann, 2009*). None of the other components demonstrated a difference between PD and controls (p > 0.05 uncorrected for all remaining components). We will call the deformation pattern showing a group difference the *PD-ICA network* from this point on. Ten of the other ICA components demonstrated an effect of age in the entire group. In three of these, there was a positive correlation between the component expression and age (meaning increased volume with age) and all three consisted of ventricle or cerebrospinal fluid space enlargement (*Figure 1—figure supplement 1*). The other seven age-related components demonstrated a negative correlation (volume loss with age) and consisted of areas of gray or white matter (*Figure 1—figure supplement 2*). The PD-ICA network also demonstrated greater atrophy with increasing age in both PD patients (r = −0.38, p < $10^{-9}$) and controls (r = −0.53, p < $10^{-9}$).

## Spatial analysis of PD-ICA network

Each spatial map was converted to a z-statistic image via a normalized mixture model and then thresholded at z ≥ 3. Regions were identified using the atlases of *Talairach and Tournoux (1988)* and *Mai et al. (2003)*. The PD-ICA network includes all components of the basal ganglia (substantia nigra, subthalamic nucleus, nucleus accumbens, putamen, caudate nucleus, and internal and external globus pallidus), the pedunculopontine nucleus, basal forebrain, including bed nucleus of the stria terminalis and an area containing the nucleus basalis of Meynert, the hypothalamus, amygdala, hippocampus, parahippocampal gyrus, and two thalamic regions, the ventrolateral nucleus and pulvinar. Cortical regions in this network are the insula, occipital cortex Brodmann area 19, superior temporal gyrus, rostral anterior cingulate cortex, premotor and supplementary areas, and parts of lateral prefrontal cortex (*Figure 1*, *Tables 1, 2*; see also *Figure 1—figure supplement 4*).

## Clinical correlation

To confirm that the PD-ICA network identified above was disease-related, we compared individual deformation values in the network to measures of disease severity. The two clinical measures used were the striatum binding ratio (SBR) measured with single photon emission computed tomography (SPECT) using the tracer [$^{123}$I]FP-CIT (*Booij and Knol, 2007*) to measure dopamine nerve terminal density, and the score on the Movement Disorder Society revised Unified Parkinson's Disease Rating Scale (UPDRS) part III (*Goetz et al., 2008*), an objective measure of motor disability. For SBR, we used the average value of left and right putamen. There was a significant correlation between individual SBR and DBM values in the PD-ICA network in the PD group (r = 0.23, p < 0.0005, *Figure 2*). This shows that the greater the loss of dopamine nerve terminals, the greater the volume loss in the PD-ICA network. There was also a significant correlation between these two measures in the control group (r = 0.33, p < 0.0005).

There was a significant correlation between DBM values within the PD-ICA network and UPDRS III in the PD patients (r = −0.22, p < 0.001; *Figure 2*). SBR was not significantly correlated with age in the PD subjects (r = −0.10, p = 0.12) but it was in controls (r = −0.35, p < 0.0001). Also, in PD subjects, SBR was significantly correlated with UPDRS III (r = 0.20, p = 0.002). We also tested whether UPDRS III was correlated with DBM values obtained from any one of the other 29 ICA components. There was only one other component marginally correlated with disease severity (r = −0.2, p = 0.048, Bonferroni corrected) consisting of the previously mentioned cerebellar network, areas VIIIA, VIIIB, CrusII, and IX. Because age, UPDRS III, and SBR all correlated with PD-ICA DBM, we performed multiple linear regressions. In the PD group, the model (PD-ICA ~ 1 + Age + UPDRS III + SBR) showed an effect of age (p = 2.4 $e^{-08}$), UPDRS III (p = 0.06) and SBR (p = 0.01). In the controls, the model (PD-ICA ~ 1 + Age + SBR) demonstrated an effect of age (p = 4.7 $e^{-08}$) but not SBR (p = 0.17).

Finally, multivariate analysis was used to look for an effect of scanning site. For each obtained DBM-network, we applied the model: DBM ~ Group (PD/Control) + Age + Gender + Site. There was no significant effect of site after correcting for multiple comparisons (p > 0.1).

## Comparing disease-related atrophy to functional networks in health

We next tested the hypothesis that the PD-ICA deformation network represents an intrinsic functional network. We compared the PD-ICA network as well as the other 29 ICA maps obtained from the DBM

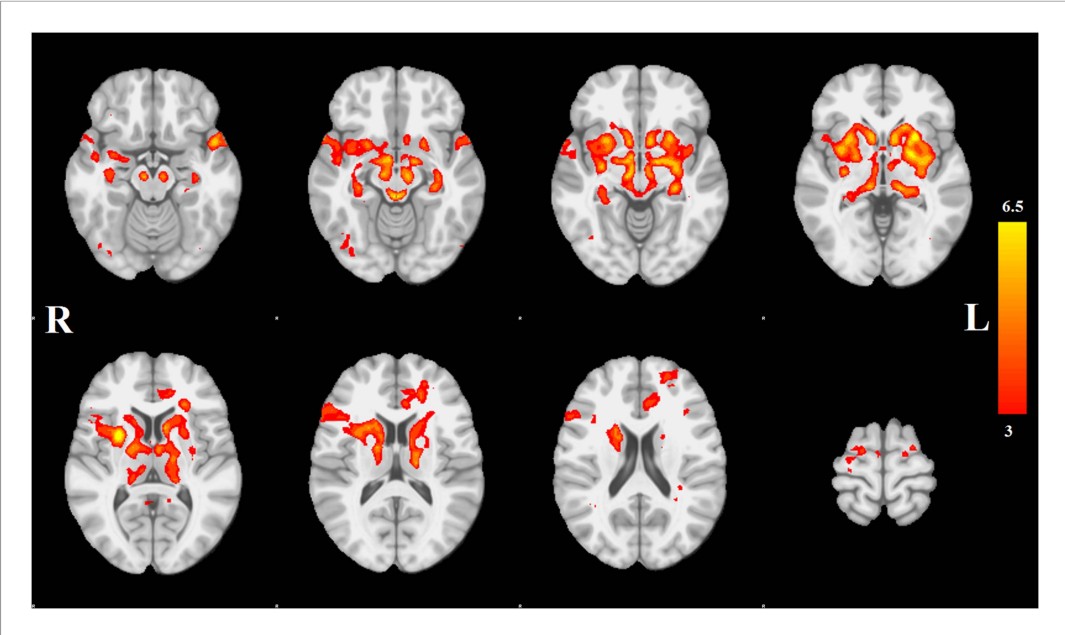

**Figure 1**. Distribution of atrophy in Parkinson's disease. This image displays the only one of the 30 independent component analysis (ICA) networks showing a significant difference between Parkinson's disease (PD) and Controls (p = 0.003 after correction for multiple comparison). The ICA spatial map was converted to a z-statistic image via a normalized mixture–model fit and then thresholded at z = 3. Selected sections in Montreal Neurological Institute (MNI) space at coordinates z = −16, z = −12, z = −7, z = −2, z = 8, z = 14, z = 20, z = 70. See *Tables 1, 2* for anatomical localization. Note that the value at each voxel is the z-score of the ICA component, not the group difference.

The following figure supplements are available for figure 1:

**Figure supplement 1**. Networks with positive correlation between average deformation and age.

**Figure supplement 2**. Networks with negative correlation between average deformation and age.

**Figure supplement 3**. Overlapping areas between the PD-ICA network obtained from the PPMI data set and regions from *Masuda-Suzukake et al. (2013)*.

**Figure supplement 4**. Voxel-wise difference in atrophy between PD and controls.

ICA analysis to intrinsic brain networks in healthy brain. In order to increase confidence in the results, we determined normal intrinsic brain networks in several different ways:

1. A seed-based functional connectivity map obtained from resting-state fMRI (rsfMRI) in 51 healthy volunteers with an a priori region of interest in the SN.
2. Two sets of 70 and 100 typical functional resting-state networks from healthy volunteers previously identified with MELODIC ICA by *Smith et al. (2009)* and *Smith et al. (2013)*.

First, we compared the resting-state SN seed-based map (*Figure 3B*) with all 30 structural maps obtained from the PPMI ICA analysis. Only two ICA networks passed the threshold of |r| > 0.25. These two networks correspond to very similar subcortical networks in basal ganglia and brainstem, one of which is mostly white matter areas (r = 0.36), while the other is mostly gray matter areas (r = 0.30). The latter is the aforementioned PD-ICA network in which atrophy correlated with disease severity (*Figure 3—figure supplement 1*).

Then, we compared the PD-ICA network to the 70 rsfMRI ICNs in normal brain provided by *Smith et al. (2009)*. One of the 70 networks passed the threshold (with r = 0.32). This network has been related to reward tasks, interoceptive functions, and motor/sensory processing (*Figure 3C*).

**Table 1**. PD-ICA subcortical anatomical areas

| Brain area | Sub-areas | Z-score L/R | Peak coordinate L/R |
|---|---|---|---|
| Entorhinal cortex | – | na (2.94)/3.1 | −20, −7, −32/19, −4, −34 |
| Claustrum | – | na/4.3 | na/36, 0, −21 |
| Amygdala | Basolateral | 3.8/4.1 | −23, −5, −21/22, −4, −20 |
| Hippocampus | Dentate Gyrus | 4.7/4.2 | −34, −18, −14/34, −15, −17 |
| Substantia nigra | – | 5/4.9 | −8, −18, −15/7, −17, −15 |
| Periaqueductal gray | – | 5.6/5.4 | −3, −34, −12/3, −33, −12 |
| Pedunculopontine nucleus | – | 4.7/4.6 | −6, −30, −11/6, −29, −11 |
| Hypothalamus | – | 3.4/4.2 | −5, −3, −11/4, −3, −11 |
| Hippocampus | CA1, CA2, CA3 | 5.3/4.5 | −30, −27, −10/31, −27, −11 |
| Subthalamic nucleus | – | 5.2/5.2 | −8, −16, −10/9, −16, −10 |
| Nucleus accumbens | – | 5/4.9 | −9, 11, −9/8, 11, −8 |
| Basal forebrain | BNST | 3.3/3.6 | −6, 4, −8/9, 3, −10 |
| Basal forebrain | Extended amygdala | 6/5.8 | −16, −6, −8/10, −6, −8 |
| Basal forebrain | Substantia innominata | 3.6/5.1 | −8, 0, −8/8, 0, −8 |
| Putamen | Anterior putamen | 5.6/4.8 | −25, 11, −5/25, 11, −5 |
| Putamen | Posterior putamen | 6.1/4.8 | −30, −12, −6/31, −15, −2 |
| Globus pallidus | Internal + external | 5.7/4.7 | −20, 1, −1/21, −3, −3 |
| Caudate nucleus | Head | 8.2/6.2 | −10, 12, 4/10, 10, 2 |
| Pulvinar | Medial/Lateral | 5.3/4.5 | −19, −31, 5/11, −26, −4 |
| Thalamus | Ventrolateral/Ventroanterior | 5.3/3.6 | −17, −14, 11/14, −14, 12 |
| Caudate | Body | 4/4.8 | −15, 11, 12/17, 10, 15 |

List of subcortical regions belonging to the PD-ICA network and their peak z-scores. (BA: Brodmann area, na: not applicable, BNST: bed nucleus of the stria terminalis, ICA: independent component analysis, PD: Parkinson's disease).

We assessed statistical significance by generating 1000 permutations of each of the 70 ICNs by reassigning the coordinates of each voxel randomly (*Figure 3—figure supplement 2*). We then repeated this comparison using a finer decomposition of 100 resting-state ICNs from the Human Connectome Project (HCP) (*Smith et al., 2013*) using MELODIC. Four components showed spatial

**Table 2**. PD-ICA cortical anatomical areas

| Brain area | Sub-areas | Z-score L/R | Peak coordinate L/R |
|---|---|---|---|
| Superior temporal gyrus | Temporal pole BA 38 | 5.6/3.6 | −50, 11, −18/50, 10, −12 |
| Occipital lobe | BA 19 | 3.1/3.4 | −39, −77, −18/35, −79, −14 |
| Insula | Mid-insula | 4.5/4.6 | −39, 0, −5/38, 5, −2 |
| Inferior frontal gyrus | BA 45 | 3.4/4.4 | −38, 26, 19/53, 26, 15 |
| Anterior cingulate cortex | Rostral ACC | 4.3/na | −6, 31, 18/na |
| Middle frontal gyrus | DLPFC BA 9/46 | 4.1/na | −22, 51, 19/na |
| Superior frontal gyrus | BA 6 | 4/3.7 | −18, −10, 66/23, −10, 54 |
| Supplementary motor area | – | na/3.4 | na/5, −12, 67 |

List of cortical regions belonging to the PD-ICA network and their peak z-scores. (BA: Brodmann area, na: not applicable, ACC: anterior cingulate cortex, DLPFC: dorsolateral prefrontal cortex, ICA: independent component analysis, PD: Parkinson's disease).

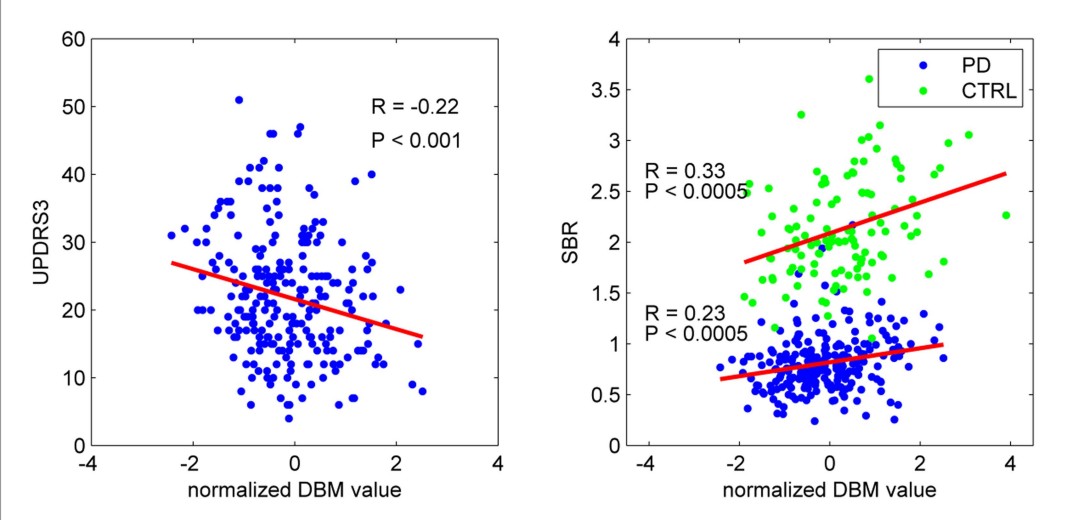

**Figure 2**. PD-ICA network, dopamine denervation, and severity of disease. Left: Unified Parkinson's Disease Rating Scale (UPDRS) part III (a measure of motor function and disease severity—higher value means more severe disease) was significantly correlated with the degree of atrophy in the network (r = −0.22, p < 0.001). Right: plot of [$^{123}$I]FP-CIT striatum binding ratio (SBR) vs deformation value in the PD-ICA (*Figure 1*). Correlation: r = 0.23, p < 0.0005 for PD patients, and r = 0.33, p < 0.0005 for age-matched controls.

overlap to the PD-ICA network using spatial correlation (*Figure 3—figure supplement 3*). The mean fMRI time series from these components were then used to determine whether they themselves belonged to one larger ICN. These time series demonstrated significant inter-correlation (p < 0.0016 by permutation testing). Finally, hierarchical clustering (*Smith et al., 2013*) confirmed that all four components clustered together.

We also compared the PD-ICA network to a map of regions responding to stimulus value during fMRI experiments as identified by meta-analysis (*Bartra et al., 2013*). In these experiments, subjects typically evaluated an offered item (say a food) and experimenters identified brain regions where the fMRI signal tracked subjective value (e.g., willingness to pay for the item). The premise is that PD may affect dopamine projection sites that encode aspects of stimulus value. Only one of the 30 networks passed the threshold (r = 0.28), namely the aforementioned PD-ICA network (*Figure 3D*).

In summary, the PD-ICA network exhibited significant spatial overlap with presumed intrinsic brain networks determined by three different methods.

## Testing the propagation model

The sequential propagation model predicts that the spatial progression of the disease process will be determined by brain network topology. Connectivity between any region and the presumed disease epicenter will determine how severely it is affected. Here, we evaluated this assumption by exploring whether the gray matter atrophy patterns observed in PD patients could be explained by functional and geodesic distance (i.e., the number of edges separating two nodes in a graph) to the hypothetical pathogenic epicenter. We chose to use the SN as the epicenter based on known PD pathology. Note that the SN is unlikely to be the first affected site in the central nervous system (CNS) (*Braak et al., 2003*, *2004*); however, we postulate that it is likely to function as a source for propagation to the supratentorial brain. Network connectivity in health can be defined functionally, using rsfMRI, or structurally, using DW-MRI. The influence of disease on each node can be estimated from the statistical difference in deformation between PD and control groups. We parcellated the brain into 112 regions of interest (ROIs, *Figure 4—figure supplement 1*) and computed the degree of deformation in each region. We also generated two connectomes from these ROI using rsfMRI and DW-MRI data from two different pools of healthy subjects. The connection strength between each pair of regions was computed as described in the 'Materials and methods'.

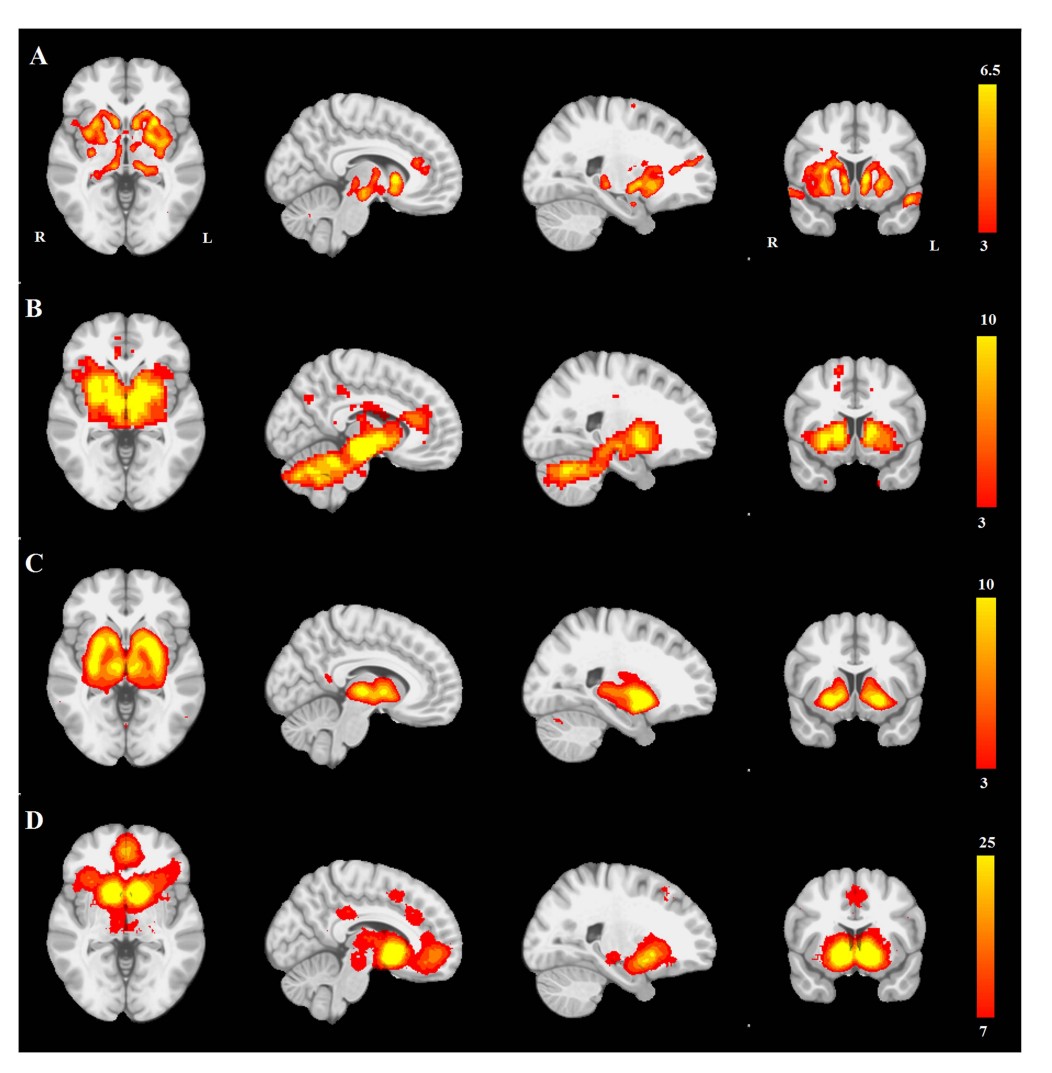

**Figure 3**. PD atrophy resembles normal intrinsic connectivity networks. Selected sections for (**A**) PD-ICA network from the Parkinson's Progression Markers Initiative (PPMI) data set thresholded at z = 3. (**B**) Seed-based resting-state functional MRI (fMRI) connectivity with substantia nigra as a priori seed. (**C**) Intrinsic connectivity network (ICN) correlated with PD-ICA from *Smith et al. (2009)*. (**D**) Regions responding to stimulus value during fMRI (meta-analysis of *Bartra et al., 2013*) (Selected slices in MNI space z = −2, x = −8, x = −23, y = 10.)

The following figure supplements are available for figure 3:

**Figure supplement 1**. Selected slices for seed-based resting-state fMRI analysis results with SN as a priori seed (top), PD-ICA network from the PPMI data set (middle), ICA network consisting of white matter areas in basal ganglia and cerebellum (bottom).

**Figure supplement 2**. The correlation between the PD-ICA network and the 70 ICNs from *Smith et al. (2009)* is displayed in red.

**Figure supplement 3**. Correspondence between the PD-ICA network and resting-state networks (RSN) from the Human Connectome Project (HCP).

There was a significant correlation between resting-state functional connectivity of each node with SN (in healthy brain) and the PD-related deformation (PD minus control t-score), (r = 0.40, p < 0.0001). We repeated the same analysis controlling for spatial proximity between each region and SN by

entering Euclidean distance as a covariate. The correlation was unchanged (r = 0.38, p < 0.0001), suggesting that the relationship cannot be explained by spatial proximity. The correlation implies that higher functional connection between a given region and SN is related to higher PD-related atrophy in that region (*Figure 4*). When using an anatomical measure of connectivity (DW-MRI) (*Figure 4*), we also observed a significant relationship between the level of atrophy of each region and its geodesic distance to the SN (r = −0.28, p < 0.003). These results were not different after controlling for Euclidean distance to SN (r = −0.25, p < 0.005). Finally, we repeated this analysis using every ROI as a potential disease propagator and found that SN was the likeliest disease propagator when using the rsfMRI connectome (*Table 3*). However, the red nucleus (r = 0.28) and subthalamic nucleus (r = 0.28) were also identified as potential propagators. Repeating this analysis using a tractography-derived connectome also revealed that the SN was one of the likeliest propagators, but numerous cerebellar regions also emerged as potential propagators (*Figure 4—source data 2*). This may be due to difficulty in accurate identification of the targets of brainstem white matter tracts using DW-MRI (*Ford et al., 2013*).

## Discussion

### Pattern of atrophy in PD

The combination of DBM and ICA of an unprecedentedly large set of MRI data at a magnetic field strength of 3T allowed us to map out the brain regions affected in de novo PD. Patients included in this study were diagnosed on average 7 months prior to the investigation and had no evidence of dementia (*Table 4*). Most MRI studies to date using T1-weighted images in PD have reported normal

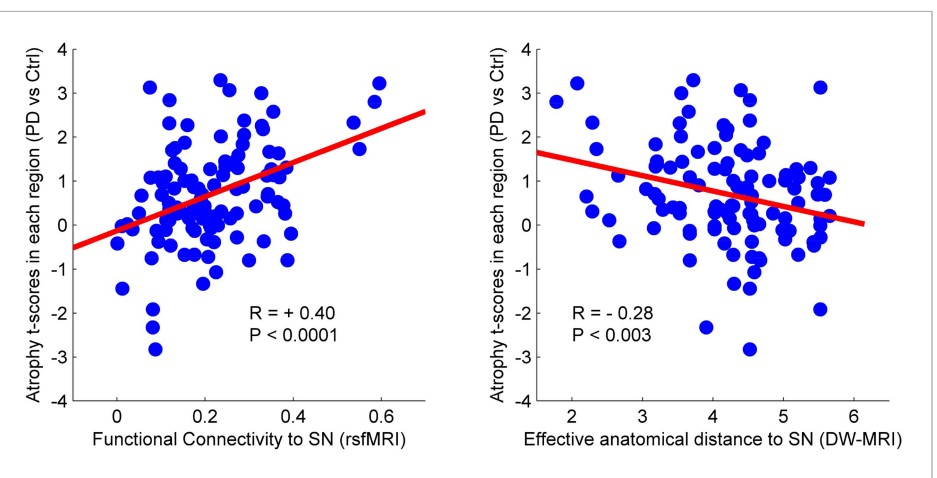

**Figure 4**. Relationship between atrophy in different brain regions and functional and structural connectivity with SN. The brain was parcellated into 112 regions (*Figure 4—figure supplement 1*). SN was chosen a priori as the region of interest, and the functional and structural connectivities between each given region and SN were calculated. The statistical difference (t-score) between the average deformation in PD and controls in each region was used as an atrophy measure. Using correlation, the relationship between regional atrophy and both regional functional connectivity with SN using resting-state fMRI (rsfMRI) (left) and regional anatomical distance using diffusion-weighted imaging (DW-MRI) (right) was examined. There was significantly greater atrophy with proximity to the SN determined functionally (r = 0.4, p < 0.0001) and anatomically (r = −0.28, p < 0.003). Note that the connectivity measure in rsfMRI is correlation, resulting in greater values for more connected regions, whereas the connectivity measure in DW-MRI is distance, resulting in smaller values for more connected regions.

The following source data and figure supplement are available for figure 4:

**Source data 1**. Atlas labels and their anatomical coordinates.

**Source data 2**. Best propagators (DW-MRI connectome).

**Figure supplement 1**. Anatomical atlas used for regional analysis.

**Table 3**. Best propagators (resting-state fMRI connectome)

| Seed region | r | Seed region | r |
|---|---|---|---|
| **Substantia nigra** | **0.40** | Cerebellum VIIIb | 0.05 |
| **Subthalamic nucleus** | **0.28** | Insula | 0.04 |
| **Red nucleus** | **0.28** | Anterior temporal lobe (lateral part) | 0.02 |
| **Cerebellum dentate** | **0.27** | Cerebellum CrusI | 0.01 |
| Pallidum | 0.23 | Superior temporal gyrus (anterior part) | 0.01 |
| Hippocampus | 0.22 | Caudate nucleus | −0.06 |
| Cerebellum vermis X | 0.21 | Superior temporal gyrus (posterior part) | −0.07 |
| Cerebellum vermis VIIIa | 0.20 | Middle and inferior temporal gyrus | −0.07 |
| Cerebellum interposed | 0.20 | Lingual gyrus | −0.08 |
| Cerebellum fastigial | 0.20 | Postcentral gyrus | −0.08 |
| Cerebellum vermis IX | 0.20 | Precentral gyrus | −0.09 |
| Cerebellum vermis VIIIb | 0.18 | Posterior temporal lobe | −0.09 |
| Cerebellum I IV | 0.18 | Inferior frontal gyrus | −0.10 |
| Cerebellum vermis VIIb | 0.17 | Middle frontal gyrus | −0.10 |
| Parahippocampal gyrus | 0.16 | Cuneus | −0.10 |
| Cerebellum V | 0.16 | Anterior cingulate gyrus | −0.12 |
| Anterior temporal lobe (medial part) | 0.15 | Occipital lobe (lateral part) | −0.12 |
| Cerebellum vermis CrusII | 0.14 | Lateral orbital gyrus | −0.16 |
| Occipitotemporal gyrus (lateral part) | 0.14 | Superior frontal gyrus | −0.16 |
| Cerebellum VIIb | 0.13 | Parietal lobe (Inferiolateral) | −0.16 |
| Cerebellum CrusII | 0.13 | Superior parietal gyrus | −0.20 |
| Cerebellum IX | 0.12 | Pre-subgenual frontal cortex | −0.20 |
| Cerebellum VI | 0.12 | Posterior orbital gyrus | −0.23 |
| Amygdala | 0.12 | Posterior cingulate gyrus | −0.23 |
| Cerebellum X | 0.11 | Medial orbital gyrus | −0.27 |
| Cerebellum vermis CrusI | 0.10 | Straight gyrus | −0.31 |
| Putamen | 0.10 | Anterior orbital gyrus | −0.33 |
| Cerebellum vermis VI | 0.08 | Subgenual frontal cortex | −0.34 |
| Cerebellum VIIIa | 0.07 | Nucleus accumbens | −0.38 |
| Thalamus | 0.06 | Subcallosal area | −0.42 |

Each brain region from the atlas was used as a potential propagator. The statistical difference (t-value) between the average deformation in PD and controls in each region was used as an atrophy measure. The correlation between this atrophy measure and the functional connectivity to the potential propagator was used as a measure of propagation strength. The potential propagator regions are sorted by correlation values.

fMRI: functional MRI, PD: Parkinson's disease.

volumes of brainstem, basal ganglia, and cerebral cortex. In the SN, measurements of volume and shape have been inconclusive, reporting either no change, a decrease, or increase in volume depending on the method used (*Pyatigorskaya et al., 2014*). However, most studies using parametric mapping have failed to report a difference in SN. Reduced putamen or caudate volume has been reported but only in advanced cases with mild cognitive impairment or dementia (*Apostolova et al., 2010*; *Silbert and Kaye, 2010*; *Pyatigorskaya et al., 2014*). Similarly, hippocampus and amygdala atrophy are typically linked to cognitive impairment (for review see *Silbert and Kaye, 2010*; *Ibarretxe-Bilbao et al., 2011*). With regard to cortical areas, studies using voxel-based morphometry (VBM), DBM, or cortical thickness measurements have reported significant differences between PD and controls in parahippocampal gyrus, inferior, middle and frontal temporal

**Table 4.** Clinical characteristics of subjects

|  | Control (n = 117) | PD (n = 232) | p value |
|---|---|---|---|
| Age (years) | 59.7 ± 11.3 | 61.2 ± 9.1 | 0.1 |
| Years of education (years) | 15.7 ± 2.91 | 15.4 ± 2.8 | NS |
| Sex (M/F/% males) | 74/43/63.2 | 155/77/66.8 | NS |
| Handedness–R/L/A | 98/11/8 | 210/17/5 | NS |
| Striatum binding ratio | 2.6 ± 0.6 | 1.4 ± 0.4 | <0.0001 |
| MoCA | 28.2 ± 1.2 | 27.3 ± 2.2 | <0.0001 |
| Disease duration (months) | – | 6.9 ± 7.1 | – |
| MDS UPDRS part III | – | 21.9 ± 9.1 | – |
| H&Y stage | – | 1.6 ± 0.5 | – |

M = male, F = female, NS = not significant, H&Y: Hoehn and Yahr, PD: Parkinson's disease, MoCA, Montreal Cognitive Assessment, UPDRS, Unified Parkinson's Disease Rating Scale. Statistical differences analyzed through an unpaired t-test or chi square test. Listed values are the mean ± standard deviation.

gyrus, parietal lobe and occipital lobe, but, once again, typically in advanced patients with cognitive impairment (*Silbert and Kaye, 2010*; *Ibarretxe-Bilbao et al., 2011*; *Camicioli, 2013*; *Pyatigorskaya et al., 2014*). Our finding of an atrophy network including brainstem and subcortical regions in these de novo patients is likely due to the larger number of participants available through the PPMI database compared to previous relatively underpowered investigations.

Eidelberg and colleagues (*Eidelberg, 2009*) were the first to use data reduction techniques (principal component analysis) to map PD pathology using [$^{18}$F]fluorodeoxyglucose (FDG) positron emission tomography. They identified a PD-related pattern (PDRP) implicating several regions identified in our analysis, namely globus pallidus, putamen, thalamus, premotor and supplementary motor areas. The expression of this PDRP differentiates patients from healthy controls and correlates with measures of disease severity. However, it probably reflects functional effects of the disease more than neuronal atrophy, as evidenced by the fact that it is normalized by levodopa or deep brain stimulation (*Eidelberg, 2009*).

## PD targets an intrinsic brain network

We found that the set of regions demonstrating atrophy in the PD group corresponded spatially to one ICN in normal brain. rsfMRI and anatomical imaging have identified consistent sets of regions that act as functional networks, by virtue of anatomical connectivity and temporal covariance of neuronal activity. We showed that our PD-ICA set of regions corresponded to a normal ICN identified by seed-based resting-state functional connectivity and ICA (*Smith et al., 2009*, *2013*). The finding that PD targets a set of connected brain regions supports the network-spread hypothesis, although an alternate explanation is that neurons in one network could share a common vulnerability. The brain regions that make up the PD-ICA network are involved in reward and motivation, as demonstrated by the fact that they respond to the subjective value of a perceived stimulus during fMRI (*Bartra et al., 2013*). All the regions of this value network receive projections from midbrain dopamine neurons, in keeping with this neurotransmitter's role in signaling incentive value (*Salamone et al., 2005*; *Berridge, 2006*).

## Network spread in PD

Recent evidence supports a hypothesis originally put forward by Braak (*Braak et al., 2003*, *2006*; *Goedert et al., 2013*), according to which pathogenic forms of the protein alpha-synuclein spread throughout the nervous system leading to a stereotypical step-wise pattern of neurological impairment in PD. Misfolded synuclein fibrils injected focally into rodent brains spread to neuronally connected but not adjacent areas (*Luk et al., 2012*; *Masuda-Suzukake et al., 2013*) (see also *Figure 1—figure supplement 3*).

Here, we tried to identify a putative best propagator or epicenter of the disease by creating functional and anatomical connectomes from an anatomical brain atlas. We showed that the SN could

act as a propagator, since the rate of atrophy in any brain region was proportional to its effective topological distance from the SN, as determined by either functional (rsfMRI) or anatomical (DW-MRI) connectivity. We note that, while the SN contains dopamine neurons whose degeneration accounts for all the key clinical features of early PD, the Braak scheme identifies the dorsal motor nucleus (DMN) of the vagus in the medulla as the first affected CNS structure. Indeed, recent evidence supports the potential transfer of pathogenic alpha-synuclein from the intestine to the DMN via the vagus nerve (*Holmqvist et al., 2014*). In the current study, the DMN did not demonstrate atrophy in PD compared to control subjects. Perhaps this is due to insensitivity of T1-weighted MRI-based anatomical methods in identifying neuronal loss in lower brainstem structures. The DMN is much smaller that the SN (approximately 1/10–1/20 in volume) and has limited contrast in T1 MRI. Our results, however, suggest that the SN could act as the propagator of the disease to the supratentorial CNS. Note, however, that spatial resolution limitations for all of the imaging modalities used here, T1-weighted MRI, fMRI, and DW-MRI, make it difficult to localize atrophy to small nuclei with complete certainty. Spatial inaccuracy in atlas generation and normalization of subject images to Montreal Neurological Institute (MNI) space may compound this problem. This is especially relevant to small structures of the basal forebrain and midbrain. For example, the propagator analysis identified SN, subthalamic nucleus, and red nucleus as potential propagators, but it is clear that the imaging techniques used here do not allow us to fully resolve these structures, either anatomically or functionally. Furthermore, because T1-weighted MRI is sensitive to iron content, changes in iron accumulation, for example, in SN, globus pallidus, or red nucleus, may be interpreted as volume changes by the DBM methodology.

Nonetheless, comparing the atrophy pattern identified here to the stages described by *Braak et al. (2003)*, *Braak et al. (2006)*, we note that all the areas identified in stage 3 (pedunculopontine nucleus, amygdala, basal forebrain, and substantia nigra), stage 4 (temporal mesocortex and hippocampus), and stage 5 (insula, cingulate cortex, and temporal neocortex) belong to our PD-ICA network. Braak hypothesized that the medial temporal lobe served as a beachhead for further propagation to the remaining cortex, and interestingly three of the best supratentorial propagators identified here were the parahippocampal gyrus, anterior medial temporal lobe, and hippocampus (*Table 3*).

Another prediction of the network-spread model is that connectome hubs should be especially vulnerable to disease spread (*Zhou et al., 2012*). Hubs are defined as brain regions that are highly connected using graph theoretical metrics such as degree (*Crossley et al., 2014*) or betweenness (*He et al., 2009*). In theory, hubs, being highly connected, should have greater exposure to a toxic agent that is spread trans-neuronally. In a recent meta-analysis, *Crossley et al. (2014)* found that most neurodegenerative diseases, including AD and fronto-temporal dementia, did indeed target hubs. However, one salient exception was PD. One possibility is that the PD studies included in the meta-analysis were too small in scale or methodologically incapable of detecting the true extent of damage. It is intriguing that our study identified numerous hub regions (*Bassett et al., 2008*; *He et al., 2009*; *Crossley et al., 2014*) in the PD-ICA network, including the medial temporal lobe, putamen, insula, occipital cortex, anterior cingulate, superior frontal gyrus, and middle frontal gyrus. One notable hub region absent from our PD-ICA network is the posterior cingulate gyrus/precuneus, an area typically affected in AD (*Buckner et al., 2009*). It would be interesting to see if this area eventually develops atrophy as PD progresses, and whether this is associated with cognitive impairment.

## Atrophy in healthy aged subjects

The PD-ICA network demonstrated a correlation between atrophy and dopamine denervation measured with SPECT, in both PD patients and age-matched control subjects. Atrophy and dopamine denervation also both correlated with age in the control group. In sum, the control subjects demonstrated age-related loss of dopamine innervation and atrophy in the PD-ICA network. This finding is also novel. It is known that healthy aging is associated with a progressive loss of dopamine neurotransmission (*Fearnley and Lees, 1991*), and that this may account for motor slowing and executive cognitive impairment that occurs with age (*Jagust, 2013*). Indeed, subtle Parkinsonian signs such as stooped posture, bradykinesia, and reduced facial expression in the healthy elderly were associated with SN neuron loss at post-mortem (*Ross et al., 2004*). Our results extend these findings by showing that neurodegeneration of the extended dopamine network also occurs in healthy aging, albeit without attaining the severity of outright PD, possibly via loss of neurotrophism, or perhaps, due

to low-level mitochondrial dysfunction or synucleinopathy (*Olanow and Brundin, 2013*). PD might then target the PD-ICA network due to a dual hit effect of pathology superimposed upon normal aging.

## Materials and methods

### Subjects and data collection

Data used in this article were primarily obtained from the PPMI database. In addition, rsfMRI and DW-MRI in healthy subjects were used to generate normal human connectomes for investigation of the network propagation model.

### PPMI data set

The PPMI is described at www.ppmi-info.org. PPMI is a public–private partnership funded by the Michael J Fox Foundation for Parkinson's Research and funding partners listed at www.ppmi-info.org/fundingpartners. It is an observational, multicenter longitudinal study designed to identify PD biomarkers (*Marek et al., 2011*). Each participating PPMI site received approval from a local research ethics committee before study initiation and obtained written informed consent from all subjects participating in the study.

For this study, we used the initial visit 3T high-resolution T1-weighted MRI scans acquired from September 2013 to January 2014. MRI data were acquired in 16 centers participating in the PPMI project, using scanners from three different manufacturers (GE medical systems, Siemens, and Philips medical systems). The acquisition parameters are detailed in the data set Website: http://www.ppmi-info.org/wp-content/uploads/2015/03/PPMI-MRI-Operations-Manual-V7-0-20JAN2015-FINAL.pdf.

We also made use of the following data for each participant: age, sex, disease duration, score on the Movement Disorder Society—UPDRS III (*Goetz et al., 2008*) while off medications, score on the Montreal Cognitive Assessment battery and striatal binding ratio (SBR) from SPECT measurements with the tracer [$^{123}$I]FP-CIT, a measure of dopamine neuron terminal density (*Booij and Knol, 2007*). In total data from 355 subjects (237 PD patients and 118 age-matched controls) were used. Six subjects, five PD patients and one age-matched control, were excluded from analysis due to failure in MRI processing. Clinical characteristics are shown in *Table 4*.

### rsfMRI

We acquired rsfMRI in 51 healthy, right-handed volunteers (mean age: 23.6 ± 5.9, range: 18–47, 32 men, 63%). None of the subjects reported a history of drug abuse, neurological or psychiatric disorder. The experimental protocol was reviewed and approved by Research Ethics Board of MNI. All subjects gave informed consent. Scans were acquired using a Siemens MAGNETOM Trio 3T MRI system at the MNI. High-resolution, T1-weighted, three-dimensional volume acquisition for anatomic localization (1-mm$^3$ voxel size) and resting-state echoplanar T2*-weighted images with blood oxygenation level-dependent (BOLD) contrast (3.5-mm isotropic voxels, TE 30 ms, TR 2 s, flip angle 90˚) were acquired from all participants. Each resting-state scan was 5-min long (150 vol). Subjects were instructed to rest quietly with eyes open.

### DW-MRI

To obtain white matter connectivity maps of normal brain, we used the Illinois Institute of Technology Human Brain Atlas v.3 (*Varentsova et al., 2014*) (http://www.nitrc.org/projects/iit2/). This atlas contains structural (T1) and high angular resolution DW-MRI data, and probabilistic gray matter maps of the adult human brain in MNI space. The atlas was generated from MRI data from 72 human subjects (42 females (59%): mean age 26.6 ± 4.8 years, range 20–39 years; 30 males: mean age 31.9 ± 4.9 years, range 22–40 years).

### DBM

Local change in tissue density was calculated using DBM. DBM consists in spatially transforming each MRI non-linearly to a stereotaxic template, and using the local deformation as a measure of tissue expansion or atrophy. There are other methods to detect population differences in brain structure. VBM measures local gray matter density by transforming the brain to stereotaxic space, segmenting

the tissue into gray and white matter and cerebrospinal fluid, and performing spatial smoothing on the gray matter maps so that local image intensity reflects gray matter density (*Ashburner and Friston, 2000*). We decided against VBM as it does not preserve the entirety of the MRI data, and there is some suggestion that it is less sensitive than DBM to subcortical atrophy (*Borghammer et al., 2010*; *Scanlon et al., 2011*). Another approach is to measure cortical thickness from the MRI (*Pereira et al., 2014*), but this would also make it impossible to detect subcortical changes.

For DBM, we registered every brain non-linearly to the MNI152-2009c template (available at http://www.bic.mni.mcgill.ca/ServicesAtlases/ICBM152NLin2009) and computed the deformation applied at each voxel using the procedure explained in *Aubert-Broche et al. (2013)*. Pre-processing of MRI included denoising using optimized non-local means filtering (*Coupe et al., 2008*), correction for intensity inhomogeneity (*Sled et al., 1998*), and linear intensity scaling using histogram matching to the MNI152-2009c template.

The resulting images were registered to MNI space using the MNI152-2009c template, in two steps: (1) A hierarchical nine-parameter linear registration was computed between native MRI images and the template by maximizing the cross correlation of intensities as the similarity measure (*Collins et al., 1994*). The resulting transformation was applied to the MR image to resample it onto a 1-mm$^3$ voxel grid and bring it into MNI space. (2) A hierarchical non-linear registration was performed on the linearly resampled scan to align it with the MNI152-2009c template (*Collins and Evans, 1997*). The resulting non-linear transformation field was inverted to generate a map of the deformations in template (MNI) space for each subject. Quality control was performed on each individual data set: the brain mask, and linear and non-linear registrations were visually inspected, and data sets with faulty registration were discarded.

After the registration procedure, for each MRI and for each position in the brain $x = x1, x2, x3$, we obtain a displacement value in each direction to generate a field of vectors: $U(x) = (u1(x), u2(x), u3(x))$; that is, during the registration procedure position, $x$ is displaced to the new position $x + U(x)$ in the template space. This shows how much each voxel was moved from the MNI152-2009c template to match the subject's brain. To estimate local atrophy, an extra step is needed. Since a completely uniform displacement results in no change of volume, we are interested in the derivative of the displacement in each direction, that is, the determinant of the local Jacobian matrix of displacement.

$$J = \frac{\partial U}{\partial x} = \begin{bmatrix} \dfrac{\partial u_1}{\partial x_1} & \dfrac{\partial u_1}{\partial x_2} & \dfrac{\partial u_1}{\partial x_3} \\ \dfrac{\partial u_2}{\partial x_1} & \dfrac{\partial u_2}{\partial x_2} & \dfrac{\partial u_2}{\partial x_3} \\ \dfrac{\partial u_3}{\partial x_1} & \dfrac{\partial u_3}{\partial x_2} & \dfrac{\partial u_3}{\partial x_3} \end{bmatrix}.$$

This Jacobian matrix is estimated using a first order approximation:

$$\frac{\partial u_3}{\partial x_2} \cong \frac{u_3(x_1, x_2 + \delta, x_3) - u_3(x_1, x_2 - \delta, x_3)}{2 \cdot \delta},$$

where $\delta$ is the voxel dimension along the $x_2$ axis. To calculate the relative ratio of the local volume change, we use the determinant of the Jacobian matrix |J| minus 1, that is, |J| − 1. By performing this calculation at each voxel, we obtain a map of local relative volumetric difference between each subject image and the MNI152-2009c template, reported in the MNI152-2009c template Talairach-like coordinate system (*Chung et al., 2001*).

## Anatomical atlas

We created a composite anatomical atlas of gray matter regions in the cerebral hemispheres, cerebellum, and midbrain. For the supratentorial regions, we used the Hammers atlas (Copyright Imperial College of Science, Technology and Medicine, Alexander Hammers and University College London 2011. All rights Reserved) (*Hammers et al., 2003*) excluding the brainstem, cerebellum, and white matter. For the cerebellum, we used a public-domain high-resolution digital cerebellar atlas (*Diedrichsen et al., 2009*). These two atlases do not have adequate segmentation of three midbrain structures, the SN, subthalamic nucleus, and red nucleus. We therefore manually segmented these three structures using the Display software tool (McConnell Brain Imaging Centre) and three sources

of anatomical information: the high-resolution MRI template (T1-weighted ICBM 2009c template, resolution = 0.5 mm³), the BigBrain (*Amunts et al., 2013*), a 20-micron resolution digital brain atlas in MNI space, and the brainstem anatomical atlas of *Duvernoy et al. (1995)*. The three structures were manually drawn on the high-resolution ICBM 2009c template. We then confirmed the segmentation of these three regions with a recently developed subcortical atlas based on ultra high-field MRI (*Keuken et al., 2014*). The composite atlas thus created contains 112 cortical and subcortical structures (*Figure 4—figure supplement 1*, *Figure 4—source data 1*) and excludes all brainstem regions caudal to the SN.

## Extracting independent components of the structural deformation

We used ICA to extract patterns of deformation in PD patients and age-matched controls. ICA is a statistical method to decompose multivariate data into statistically independent components (*Calhoun et al., 2001*; *Beckmann and Smith, 2004*; *Hyvärinen et al., 2004*). In this case, we used ICA to decompose the deformation maps into spatially distinct subcomponents. ICA was performed with MELODIC (http://fsl.fmrib.ox.ac.uk/fsl/fslwiki/MELODIC), a toolbox that is part of the FSL analysis package (*Beckmann and Smith, 2004*, *2005*; *Smith et al., 2004*; *Douaud et al., 2014*). All the individual DBM images (patients and controls combined) were concatenated to create a 4-D image in which the first three dimensions are the individual 3-D deformation maps and the fourth dimension consists of the subjects. MELODIC applies an initial principal component analysis-based dimension estimation to find the optimal number of independent components and then uses this number in the decomposition procedure to identify the independent spatial components. Each of these components consists of a vector of normalized DBM values (one per subject) as well as a corresponding 3-D spatial map. The spatial maps were converted to z-statistic images via normalized mixture-model fitting, and thresholded at $z = 3$ (*Beckmann and Smith, 2004*; *Smith et al., 2009*). The ICA algorithm in MELODIC is sensitive to sparsely distributed (super-Gaussian) data (*Daubechies et al., 2009*) as typically seen in fMRI. We confirmed that the DBM data used here possessed this super-Gaussian property (kurtosis >4).

## Statistical analysis of independent networks

To identify regions showing greater atrophy in PD, the average DBM values for each ICA component and each subject were entered in an unpaired t-test comparing PD subjects and age-matched controls (Bonferroni corrected for multiple comparisons). Also, the DBM values from each component were correlated with the age of each subject to identify patterns of deformation associated with aging. Again, Bonferroni corrections were applied. The DBM values from the component(s) that demonstrated a statistically significant group difference were compared to two disease-related clinical measures using linear regression: SBR and UPDRS III.

## rsfMRI analysis

In order to test the theory that PD targets normal brain networks, we analyzed rsfMRI data from 51 young healthy individuals. The rsfMRI data were preprocessed using the Neuroimaging Analysis Kit (NIAK) (*Bellec et al., 2010*, *2012*), to perform slice timing correction, rigid body motion correction, and removal of slow temporal drift using a high-pass filter with 0.01 Hz cut-off (*Perlbarg et al., 2007*). Physiological noise was accounted for by including white matter and cerebrospinal fluid signals as covariates. In the next step, the mean motion-corrected volume of each subject's fMRI data was first linearly (6 parameters: 3 translations, 3 rotations) registered to the native individual T1 image and then non-linearly registered to the MNI152 non-linear template (http://www.bic.mni.mcgill.ca/ServicesAtlases/HomePage). All data were resampled and smoothed with a 6-mm Gaussian kernel. All fMRI time series further underwent level one auto-regression (AR1) temporal de-noising.

The mask of the SN described above was used to extract the average BOLD time series of each individual scan. The time series from right and left SN were averaged. Seed-based functional connectivity analysis was performed (*Worsley et al., 2002*; *Worsley, 2005*) as implemented in fmristat (http://www.math.mcgill.ca/keith/fmristat/). The SN average BOLD signal was entered as a regressor in the design matrix and its correlation calculated with all voxels in the brain. A mixed effects model was applied to generate a t-map for the group. This was thresholded based on random field theory to achieve a p value of 0.05 after correction for multiple comparisons.

## Comparing PD-related structural atrophy networks and functional networks in healthy brain

We quantified the spatial similarity between PD-ICA atrophy maps and intrinsic networks in the healthy brain. We compared the PD-ICA map to intrinsic brain networks obtained from three different sources: the functional connectivity map obtained from the SN seed region, and ICNs from two published sources (*Smith et al., 2009*, *2013*). We also compared the PD-ICA network map to a map derived from a meta-analysis of fMRI experiments where subjects tracked the value of rewarding stimuli (*Bartra et al., 2013*).

First, the functional connectivity map obtained from the SN seed-based analysis was compared to the 30 different ICA maps from the deformation analysis. Second, we compared the PD atrophy map (PD-ICA network) to the 70 ICNs identified by Smith et al. based on ICA analysis of resting-state data in healthy subjects (*Smith et al., 2009*). We further compared the PD-ICA map to 100 ICNs identified from the HCP rsfMRI data (*Smith et al., 2013*). Then, we compared the 30 ICA maps from the deformation analysis to a map identified in a meta-analysis of value-related fMRI studies, reasoning that dopamine networks implicated in PD would be similar to value networks in healthy brains (i.e., regions where BOLD signal tracks the value of experimental stimuli). Spatial cross-correlation (Pearson) was used as the measure of similarity. We chose $|r| > 0.25$ as indicative of similarity between two spatial maps as this has been argued to guard against false positives in a similar comparison between ICA-derived spatial maps (*Smith et al., 2009*).

## Relating deformation to the normal connectome

We generated the normal connectome of our 112 region brain atlas (52 paired bilateral regions, 8 midline regions), in two different ways, with rsfMRI and with DW-MRI. The goal here was to test the theory that geodesic (synaptic) proximity to the SN in healthy brain would predict the distribution of regional atrophy in PD.

## Generation of the functional connectome

The modified Hammers-Cerebellum-Brainstem atlas was used to extract the average BOLD time-series, after correction for physiological noise, for all regions for each of the 51 rsfMRI acquisitions. All 112 regions in the atlas were used as separate masks and average time series were extracted for each. The time series used were the residuals of the linear model after correction for physiological noise and head motion. As a result, we obtained the correlation between all region pairs, which gives a $112 \times 112$ connectivity matrix for each acquisition. Then, a common group-based matrix called $R_{group}$ was calculated. To do so, we followed steps explained in *Carbonell et al. (2014)*: each individual subject's (j) correlation matrix $R_{ind}^{j}$ was converted to a standard normal metric using the Fisher transformation

$$Z_{ind}^{j} = \frac{1}{2} \log\left(\frac{1 + R_{ind}^{j}}{1 - R_{ind}^{j}}\right).$$

Then, all the Fisher transformed results were averaged.

$$Z_{group} = \frac{\sum_{j=1}^{N} Z_{ind}^{j}}{N},$$

where N is the number of subjects. Finally, the group Z correlation matrix result was converted back to the correlation space using an inverse Fisher transform

$$R_{group} = \frac{e^{2Z_{group}} - 1}{e^{2Z_{group}} + 1}.$$

The modified Hammers-Suit-Brainstem atlas was also used to extract the average deformation values (Jacobian) for each region for each PD and age-matched control subject. A t-test was performed in each region to assess the difference between the PD and control groups, corrected for age. These t-values are a measure of the difference between the two groups and the sign of the t-value shows the direction of the effect, with negative values chosen to mean greater deformation (atrophy) in PD.

The relationship between functional connectivity of each brain region with the selected seed region (SN) and the t-value of the deformation in each region (PD minus age-matched controls) was investigated using correlation analysis.

### Generation of the DW-MRI connectome

Probabilistic anatomical connectivity values between each pair of atlas regions were estimated using a fully automated fiber tractography algorithm (*Iturria-Medina et al., 2007*) and the intravoxel fiber orientation distribution functions from the IIT Human Brain Atlas v.3 (*Varentsova et al., 2014*). A maximum of 500-mm trace length and a curvature threshold of $\pm 90°$ were imposed as tracking parameters. Based on the resulting voxel-region connectivity maps, the anatomical connection probability (ACP) between any pair of regions i and j ($ACP_{ij} = ACP_{ji}$) was calculated as the maximum voxel-region connectivity value between both regions. The ACP measure (*Iturria-Medina et al., 2007*) reflects the degree of evidence supporting the existence of each hypothetical white matter connection, independently of the density/strength of this connection, and is thus a measure of low susceptibility to gross fiber degeneration related to the aging processes. Then, effective anatomical distances between each region i and all the other ROI were estimated as the lengths of the shortest paths (in terms of ACP values) linking that region to all the other regions (*Iturria-Medina et al., 2014*).

Then, as for the functionally derived connectome above, we tested whether the PD minus control atrophy value in each atlas region depended on the geodesic distance to the SN in the DW-MRI connectome. We calculated the non-linear correlation between the regional T value of the difference in Jacobian between patients and controls and the effective anatomical distance to the SN.

## Acknowledgements

This work was funded by grants from the Michael J Fox Foundation for Parkinson's Research, the W Garfield Weston Foundation, and the Alzheimer's Association, the Canadian Institutes for Health Research, and the Natural Sciences and Engineering Research Council of Canada. We thank Christian Beckmann and Simon Eickhoff for their advice on data analysis. Data used in this article were obtained from the Parkinsons Progression Markers Initiative (PPMI) database (www.ppmi-info.org/data). For up-to-date information on the study, visit www.ppmi-info.org. PPMI is sponsored and partially funded by the Michael J Fox Foundation for Parkinsons Research and funding partners, including AbbVie, Avid Radiopharmaceuticals, Biogen, Bristol-Myers Squibb, Covance, GE Healthcare, Genentech, GlaxoSmithKline (GSK), Eli Lilly and Company, Lundbeck, Merck, Meso Scale Discovery (MSD), Pfizer, Piramal Imaging, Roche, Servier, and UCB (www.ppmi-info.org/fundingpartners). Some data used in this paper were also provided by the Human Connectome Project, WU-Minn Consortium (Principal Investigators: David Van Essen and Kamil Ugurbil; 1U54MH091657) funded by the 16 NIH Institutes and Centers that support the NIH Blueprint for Neuroscience Research; and by the McDonnell Center for Systems Neuroscience at Washington University.

## Additional information

### Funding

| Funder | Grant reference | Author |
|---|---|---|
| Michael J. Fox Foundation for Parkinson's Research (MJFF) | 320897 | D Louis Collins, Alain Dagher |
| W. Garfield Weston Foundation | 320897 | D Louis Collins, Alain Dagher |
| Alzheimer's Association | 320897 | D Louis Collins, Alain Dagher |
| Canadian Institutes of Health Research (Instituts de recherche en santé du Canada) | MOP-136776 | Alain Dagher |
| Natural Sciences and Engineering Research Council of Canada (Conseil de Recherches en Sciences Naturelles et en Génie du Canada) | 436259-13 | Alain Dagher |

The funders had no role in study design, data collection and interpretation, or the decision to submit the work for publication.

## Author contributions
YZ, MU, DLC, Conception and design, Analysis and interpretation of data, Drafting or revising the article; YI-M, MD, YZ, KM-HL, VF, ACE, Analysis and interpretation of data, Drafting or revising the article; AD, Conception and design, Acquisition of data, Analysis and interpretation of data, Drafting or revising the article

## Author ORCIDs
Alain Dagher, http://orcid.org/0000-0002-0945-5779

## Ethics
Human subjects: For the Parkinson's Progression Markers Initiative (PPMI) database (www.ppmi-info. org/data). Each participating PPMI site received approval from a local research ethics committee before study initiation, and obtained written informed consent from all subjects participating in the study. For the resting state fMRI data collected in our lab, We acquired resting state fMRI in 51 healthy, right-handed volunteers. The experimental protocol was reviewed and approved by Research Ethics Board of Montreal Neurological Institute. All subjects gave informed consent.

# Additional files

## Major datasets
The following dataset was generated:

| Author(s) | Year | Dataset title | Dataset ID and/or URL | Database, license, and accessibility information |
|---|---|---|---|---|
| Zeighami Y, Ulla M, Iturria-Medina Y, Dadar M, Zhang Y, Larcher KM-HL, Fonov V, Evans AC, Collins DL, Dagher A | 2015 | Network structure of brain atrophy in de novo Parkinson's Disease | http://neurovault.org/collections/860/ | Publicly available at NeuroVault (collection 860). |

The following previously published datasets were used:

| Author(s) | Year | Dataset title | Dataset ID and/or URL | Database, license, and accessibility information |
|---|---|---|---|---|
| Marek K et al, | 2011 | Parkinson Progression Marker Initiative (PPMI) | http://www.ppmi-info.org/ | PPMI data and specimens are made accessible through the Web site to academic and industry researchers to perform verification studies of PD biomarkers. Investigators are required to submit basic information about themselves for basic administrative review to ensure legitimacy. Upon approval, investigators will be given immediate access. |

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
