## [Decision Letter]

Thank you for submitting your work entitled “Parkinson's disease targets an intrinsic brain network” for peer review at *eLife*. Your submission has been favorably evaluated by Timothy Behrens (Senior Editor), a Reviewing Editor, and two reviewers.

The reviewers have discussed the reviews with one another and the Reviewing editor has drafted this decision to help you prepare a revised submission.

This study is a very interesting and well thought out multimodal analysis of brain atrophy and functional connectivity in a multi-site cohort of de novo PD patients. It benefits from a (multi-centric) large cohort of rare de novo PD patients and from the methodological approaches used for the analysis of local atrophy (namely DBM+ICA). The fact that the authors also looked for a spread of the disease via structural and functional networks (using both diffusion imaging and [resting-state] fMRI) is another strength of this study. It provides intriguing evidence in support of the hypothesis that atrophy initiated in or near the substantia nigra may spread to other subcortical and cortical regions in a pattern that at least to some degree reflects network connectivity as revealed by resting-state fMRI.

Both reviewers were enthusiastic about many aspects of the study, but both raised substantial concerns that are largely complementary to one another. Hence, we consider the manuscript to be potentially acceptable, but only after a major revision that addresses the key points raised by the reviewers.

Some of the recommendations would entail significant additional analysis, and we do not insist that all of them necessarily be carried out. For example, suggestion #8 is to carry out the rsfMRI analysis of functional connectivity using the freely available HCP dataset. We encourage you to consider this, but it is not essential for the revised manuscript. For the other recommended re-analyses, including #1 (seed-based analysis), #11 (higher-dimensional ICA decomposition) and #13 (FLICA analysis) it is important that you either follow the recommendation or provide a cogent response as to why this was not done.

Essential revisions:

1) A strength of this paper is that multiple different methods were used to measure connectivity patterns in normal subjects (resting state seed based, resting state ICA, DTI). A relative weakness is that only one method was used to define the atrophy pattern in PD, the central finding in the paper. There are numerous techniques for detecting and quantifying atrophy, so why did the authors choose the one they did (DBS). Do they get similar results using an alternative technique? Similarly, they only used one method to identify the atrophy pattern (ICA). Why not use a seed based approach to identify regions whose atrophy correlates with atrophy in the SN? The authors do not need to perform every methodological possibility, but the reasons for their choices need to be more clearly justified. Further, they should make it clear when and why their methods deviate from prior work with similar goals (e.g. Bill Seeley's work).

2) Results are a bit overstated at times which could detract from the importance of the findings. The authors convincingly show that a specific pattern of atrophy is related to PD, aging, dopamine binding in the striatum, and UPDRS score. This alone is very worthy of publication. Whether this atrophy network is an “intrinsic brain network”, as defined by resting state fcMRI, or validates the “network spread model” of PD, this is indeed an important question, but it is weakly supported by the present data. The authors may be better served to focus on their strongest findings and relegate the others to the Discussion.

3) The authors put great emphasis on the fact that their atrophy pattern matches an “intrinsic connectivity network”, including making this the title, but the data supporting this claim are weak. Specifically, the criteria for a “match” are arbitrary. The authors chose a threshold of r = 0.25. If they had chosen a threshold of 0.35 instead, we would conclude that the atrophy pattern fails to match any intrinsic connectivity networks. Rather than concluding that the atrophy matches or fails to match an intrinsic connectivity pattern, the authors could make better claims on comparative matching. In other words, they can claim that their atrophy pattern matches a specific network better than other atrophy patterns and they can conclude that their atrophy pattern matches a specific network better than other networks.

4) It is a bit unclear whether the PD-ICA network (Figure 1) shows the full ICA component identified combining PD and controls or if only those voxels within the component that showed significant differences between PD and controls. I believe it's the former, but this should be made a bit clearer and it would be helpful to also show that latter. What part of this network shows the greatest difference between PD and controls?

5) There are concerns regarding the correlations across the 135 ROIs. The authors already have atrophy and connectivity measures at the voxel level so why not do the analysis across voxels? By combining different pieces of various brain atlases with their own hand-drawn atlas of brainstem structures, the authors introduce the possibility of bias into their ROI analysis. Is there no suitable existing atlas such as the WFU-Pickatlas? If the authors must use a custom atlas, some criteria regarding which brainstem structures were included versus excluded are needed.

6) Atrophy in the PD-ICA, SBR, UPDRS, and age all appear to be somewhat correlated. It would be interesting to know which are independently correlated after accounting for the others using a multivariate analysis.

7) In testing whether the PD-ICA overlaps with an intrinsic connectivity network, the authors include comparison to a meta-analysis of regions responding to stimulus value. Although potentially interesting, this map should not be referred to as an “intrinsic connectivity network”.

8) The resolution used for the rsfMRI connectivity analyses in healthy young subjects (both seed-based and “propagation model”) is of 3.5 mm isotropic, which makes it impossible to distinguish (a seed in) the substantia nigra from the subthalamic nucleus, and probably also the red nucleus. This might explain why the authors found that the latter two structures were as likely to be propagators as the substantia nigra.

To alleviate these major concerns, the authors should probably re-do these analyses with an improved resolution dataset, which is for instance readily available in the HCP in a young and healthy population (∼500 subjects at 2 mm isotropic).

9) The authors should explain clearly how they manually defined their ROI in the substantia nigra and other small structures (only names of anatomical atlases are specified in the Methods), and extensively discuss the inherent limitations coming with such a resolution for both their seed-based analysis and propagation model.

10) Similarly, in the subsection “Spatial Analysis of PD-ICA network”, it is not clear that the location of the T1 weighted results obtained from DBM can be so precisely identified (substantia nigra vs. subthalamic nucleus, PPN, bed nucleus of the stria terminalis, etc.), so the authors should make this point clear in their manuscript.

11) The spatial cross-correlation between the 3 different networks seems to some extent arbitrarily set up at |r|>0.25. The authors should possibly report whether other significant cross-correlations were found for |r|<=0.25. The authors might also want to use the higher dimension ICA decomposition (d=70 instead of d=20) provided in [66], as their high dimension ICA yields more specific basal ganglia networks.

12) Regarding the DBM analysis, the use of ICA is quite ingenious, especially considering the multi-centric aspect (16 sites, 3 different scanners) of this imaging cohort. Could the authors please specify whether they found site-specific or scanner-specific artefactual ICs in their results? What about a direct comparison of their DBM maps between the 2 populations? Presumably, this provided no significant result, which therefore sends a strong methodological message about an optimised approach for multi-centric T1 weighted volumetric studies.

13) There is concern about the specific use of MELODIC on structural data. The reason is that MELODIC is “tuned” to identifying sparse data and is inherently more suited for fMRI rather than structural data. It can therefore be the case that some more “global” components explaining the largest variance across the subjects can be missed sometimes. It seems unlikely here as the inputs used for TICA were DBM maps and not GM maps, and the main IC is reassuringly highly relevant to the pathology studied. It might be worth however for the authors to run FLICA (another data-driven ICA tool available in FSL) on their data to make sure results are similar.

14) Regarding their correlations with clinical measures, could the authors maybe justify in the manuscript why they have not used the MoCA and other parts of the UPDRS than part III, or alternatively carry out these correlation analyses?

15) Could the authors please provide the list of best propagators using diffusion imaging (similar to Table 3)? It would also be interesting to discuss the strong negative correlations reported in Table 3.

---

## [Author Response]

*[…] Some of the recommendations would entail significant additional analysis, and we do not insist that all of them necessarily be carried out. For example, suggestion #10 is to carry out the rsfMRI analysis of functional connectivity using the freely available HCP dataset. We encourage you to consider this, but is not essential for the revised manuscript. For the other recommended re-analyses, including #1 (seed-based analysis), #11 (higher-dimensional ICA decomposition) and #13 (FLICA analysis) it is important that you either follow the recommendation or provide a cogent response as to why this was not done*.

Here is a quick summary of the changes made. All the suggested analyses have been attempted, and most are now included:

1) Seed based structural covariance was carried out. Results were similar but possibly less informative than ICA.

2) HCP data used:

a) For rsfMRI connectome, the results very similar, but HCP data suffered from subcortical signal loss when corrected for physiological noise with the FIX algorithm;

b) HCP rsfMRI data were used for more fine-grained ICA decomposition (N=100).

3) Higher dimensional decomposition performed at 70 and 100 components.

4) Greater justification for the claim that atrophy network matches an intrinsic brain network:

a) Finer decompositions were used.

b) Permutation testing was carried out to guard against false positives.

5) FLICA was tried, but we found to be less useful than MELODIC for DBM data. We also confirmed that DBM data have the super-Gaussian distribution (high kurtosis) required by FastICA algorithm used in MELODIC.

6) We have changed the title in accordance with the change of emphasis from proving the network hypothesis to mapping atrophy in de novo PD.

7) Subjectivity and bias from homemade brain atlas were mitigated by using three published atlases for ROI delineation (this doesn’t rule out error, but reduces the possibility of observer bias).

8) Multiple linear regression for clinical scores (age, UPDRS, SBR) were carried out.

9) Tractography-based connectome was added to the resting state fMRI connectome for the epicenter analysis.

10) Voxelwise epicenter analysis was also carried out.

*Essential revisions*:

1) A strength of this paper is that multiple different methods were used to measure connectivity patterns in normal subjects (resting state seed based, resting state ICA, DTI). A relative weakness is that only one method was used to define the atrophy pattern in PD, the central finding in the paper. There are numerous techniques for detecting and quantifying atrophy, so why did the authors choose the one they did (DBS). Do they get similar results using an alternative technique?

There are essentially three commonly used data-driven methods to detect brain atrophy: Deformation based morphometry (DBM) as used here, voxel based morphometry (VBM), and cortical thickness measurements. It is also possible to automatically or manually segment the brain into anatomical structures and estimate their size, but we were interested in a whole brain parametric mapping approach for use with ICA.

We were especially interested in the subcortical and brainstem structures, given the evidence from post-mortem studies that these are the earliest targets in PD. This ruled out cortical thickness analysis.

VBM uses segmentation of the brain into grey matter, white matter and CSF followed by low-dimensional spatial registration of the grey matter map to a template. The variable used in parametric analysis is the percent grey matter in each voxel. It has the advantage of being fairly robust and easy to implement. Disadvantages include the fact that both changes in MR signal intensity and tissue density may equally affect VBM measurements, and that only data labelled as grey matter are used in the analysis (i.e. not all the MR image data are used). VBM does not preserve shape information. VBM also requires considerable spatial smoothing (to control for inter-individual cortical variability), to generate maps in which intensity reflects tissue grey matter density. Smoothing reduces spatial resolution. Also it renders the data more Gaussian, making it less suitable for MELODIC (see below). Most importantly however, VBM does poorly in the brainstem and subcortical areas.

DBM retains all of the MRI data, and uses high-dimensional nonlinear registration to the template. It is sensitive to local shape changes due to atrophy. The main disadvantage of DBM lies in the complexity of the method and requirement for accurate quality control (QC). Our group has implemented and validated a DBM pipeline method that easily allows trained users to perform QC on every scan.

There are few studies directly comparing DBM to VBM. Scanlon et al. (61) showed that DBM was superior to VBM in detecting subtle subcortical abnormalities in temporal lobe epilepsy. Borghammer et al. in a pilot study found that DBM might be superior to VBM in detecting atrophy in PD (12). In sum, DBM was used because the preliminary evidence supports its use to detect subcortical atrophy, and because it retains all of the MR information and does not require additional spatial smoothing, making it preferable for ICA analysis. We used a validated pipeline (3) and performed extensive QC (every registration verified visually).

We investigated the correlation between DBM and VBM in different areas of the brain, in the PPMI dataset. In cortical areas as well as cerebellum there is a high degree of similarity between the two methods; however in subcortical areas and brainstem VBM appears to be less sensitive than DBM (and not sensitive at all in the Brainstem). We compared the t-value of the difference between PD and control at each voxel for the two procedures. The correlation between voxelwise DBM and VBM t-values for whole brain is r = 0.54, (p<.001), the correlation for cerebellum and cortical areas are r = 0.65 and r = 0.6, respectively. For Basal ganglia the correlation is r = 0.46 and for brainstem there is no significant correlation between the two measures (r = -0.01, p = 0.26). The graphs below (Figure 5) show that there is less variability in t-values for brainstem and basal ganglia for VBM than DBM. This supports the notion that VBM is relatively insensitive to atrophy in subcortical areas. A new paragraph detailing this was also added to the Methods section (“DBM consists in spatially transforming each MRI […] make it impossible to detect subcortical changes”).

Author response image 1.**DOI:**
http://dx.doi.org/10.7554/eLife.08440.023

*Similarly, they only used one method to identify the atrophy pattern (ICA). Why not use a seed based approach to identify regions whose atrophy correlates with atrophy in the SN? The authors do not need to perform every methodological possibility, but the reasons for their choices need to be more clearly justified*.

ICA has two advantages over seed-based methods: (1) it is data driven and hence not limited by a-priori hypotheses; (2) data-reduction increases the ability to detect changes. Also it has the power to distinguish independent sub-components that might be considered part of the same map using seed based analysis.

We performed the seed based analysis of the DBM maps, with SN as a seed region. (This looks for areas that show structural covariance with the seed across the group.) This results in a map (see Figure 6, panel A, thresholded at t=3.0) that is very similar to our ICA result. The PD group has significantly greater atrophy than controls within the map (t=2.04, p=0.04). However, with ICA, there are two components (Figure 6, panel B, coloured in red and blue) that overlap with the seed-based map: (1) the PD-ICA (Figure 6, panel C) and (2), a component that consists of cerebellum and white matter near the basal ganglia (Figure 6, panel D). Deformation in the first ICA differed between PD and controls (t=3, p=0.003), however, there was no significant difference in the second ICA (t=1.2, p = 0.20). This suggests that the seed-based analysis yields somewhat similar but less specific results that may include components not specifically PD related.

Author response image 2.**DOI:**
http://dx.doi.org/10.7554/eLife.08440.024

We also tried the more traditional univariate approach: comparing DBM values at each voxel. The image is show here (Figure 7) with thresholded at z=2.5:

Author response image 3.**DOI:**
http://dx.doi.org/10.7554/eLife.08440.025

Only the substantia nigra and small peaks in basal ganglia in medial temporal lobe survive rigorous statistical thresholding. The univariate approach suffers from low power.

*Further, they should make it clear when and why their methods deviate from prior work with similar goals (e.g. Bill Seeley's work)*.

A few groups have attempted to relate brain atrophy to neurodegeneration using a network-based framework/approach. The Seeley group published two papers on Alzheimer’s disease (AD) and frontotemporal dementia (FTD) that were a major influence on our current work. The ideas of (1) comparing a disease atrophy map to a normal resting state (62) and (2) using graph theory to identify an epicenter for disease spread (75), both of which support the network spread hypothesis, originated from these two papers.

In the first they showed that atrophy maps from 5 different dementing syndromes each showed overlap with ICNs constructed using seed-based connectivity mapping (the seeds selected from the atrophy maps). This is similar to what we did here; they compared two maps: the atrophy statistical map for each disease and an ICN seeded from the most affected voxel of that atrophy map (we used completely data-driven ICNs). They used a Goodness of Fit measure between these ICNs and a binarized version of the atrophy map. We think our approach is quite similar to theirs except that because they had 5 diseases, they were able to show that the ICN generated from disease *i* fit the atrophy pattern from disease *i* better than that from the other 4 diseases. Strictly speaking this does not prove that the atrophy map is a resting state network. We went somewhat further to demonstrate that the PD atrophy map corresponded to a set of connected brain regions.

Other papers have attempted to relate ICA components from different types of data, as we have done here ([66]; Segall et al., 2012). As discussed below these papers used the same approach as us, by setting a somewhat arbitrary threshold for spatial correlation. Another recent paper (28) compared ICA components obtained in normal aging (using VBM rather than DBM) to atrophy maps from AD and schizophrenia. They also used spatial correlation but tried to assess statistical significance by generating 1000 random maps and demonstrating that the correlations between the true maps were outside the confidence interval of the correlations to the random maps. We have now implemented a version of this approach as well (Figure 3—figure supplement 2).

Finally, two recent papers have used graph theory and epidemiological spread models to support network spread models in AD (58; 43). We similarly used graph theory to identify a supposed disease epicenter, but both papers pushed this approach further in ways that are beyond our current work.

*2) Results are a bit overstated at times which could detract from the importance of the findings. The authors convincingly show that a specific pattern of atrophy is related to PD, aging, dopamine binding in the striatum, and UPDRS score. This alone is very worthy of publication. Whether this atrophy network is an “intrinsic brain network” as defined by resting state fcMRI or validates the “network spread model” of PD are important questions, but more weakly supported by the present data. The authors may be better served to focus on their strongest findings and relegate the others to the Discussion*.

We now temper our claims regarding evidence of PD as a “nexopathy”, and rather emphasize the use of ICA and DBM to map atrophy in early PD.

For example the Results section originally started with “We also tested the hypothesis that the pattern of neurodegeneration in PD is consistent with spread of a pathogen through intrinsic brain networks…”. It now starts with the deformation-based morphometry results. The Introduction has also been reconfigured.

The Abstract, which originally started with: “We tested the network-spread model of neuro-degeneration in Parkinson’s Disease (PD) using MRI and clinical data…”, now begins with*:* “We mapped the distribution of atrophy in Parkinson’s Disease (PD) using MRI and clinical data…”.

The title has been changed to “Network Structure of Brain Atrophy in de novo Parkinson’s Disease”. The word “network” only refers to the fact that we use a brain network framework in the analysis.

In sum, the atrophy findings are emphasized, and the comparison to intrinsic networks secondary.

*3) The authors put great emphasis on the fact that their atrophy pattern matches an “intrinsic connectivity network”, including making this the title, but the data supporting this claim are weak. Specifically, the criteria for a “match” are arbitrary. The authors chose a threshold of r = 0.25. If they had chosen a threshold of 0.35 instead, we would conclude that the atrophy pattern fails to match any intrinsic connectivity networks. Rather than concluding that the atrophy matches or fails to match an intrinsic connectivity pattern, the authors could make better claims on comparative matching. In other words, they can claim that their atrophy pattern matches a specific network better than other atrophy patterns and they can conclude that their atrophy pattern matches a specific network better than other networks*.

We now use the finer decomposition (N=70 and N=100). We show that there are other ICNs that correlate with the atrophy map, but that these ICNs are themselves inter-connected (see Figure 3—figure supplement 3).

However, the question “is the set of PD affected regions an intrinsic network in healthy brain?” is perhaps not answerable unequivocally in a statistical sense. (Note however that the epicenter analysis in our paper showing that PD related atrophy is predicted by distance from the SN in the normal connectome also fits with the network spread hypothesis.)

We agree that the threshold choice is to a certain extent arbitrary. It was based on (66) and (Segall et al., 2012). These authors used Pearson correlation with |r|<=0.25 and |r|<=0.20 respectively, to look at the correspondence between sets of ICA-derived networks. In both cases the threshold was chosen based on the estimated degrees of freedom (number of resels) in the images. Smith et al. calculated that |r|<=0.25 effectively protected against multiple comparisons in a dataset that, like ours, compared two sets of ICA networks derived from MELODIC (one of the two sets being the 70 network decomposition used here). If one accepts the [66] argument, then |r|<=0.25 protects against false positives for these data.

We add two other pieces of evidence. First, to show that the correlation between the PD atrophy map and the resting-state ICN exceeds chance levels we generated 1000 sets of 70 ICNs by randomly reassigning voxel values in each ICN (i.e. keeping the same z-scores but reassigning them spatially). We then calculated the spatial correlation between the PD atrophy map and each of the 70 maps thus generated, 1000 times. In the figure below we plot the correlation between the PD atrophy map and the true ICNs from the 70-network decomposition in red, and the confidence interval for the correlations with the random networks. This shows that that in the true 70 ICN decomposition one network is more significantly correlated than the others, and that this pattern is significantly above chance levels.

Note that the higher dimension ICA decomposition (d=70 instead of d=20) provided in [66] is used as suggested by the reviewers (in point #11). This in fact improved the similarity measure between the network of interest (PD-ICA) and the most similar network from r=0.28 to r=0.32.

Note that with this finer decomposition there are three other networks showing correlation (although less than 0.15). To test if these are themselves correlated we used a different rsfMRI decomposition: the 100 ICN decomposition from the human connectome project (67). We identified four networks that correlated spatially with the PD-ICA network. (We could not do this with the original decomposition from [66] because the BOLD timeseries are not available.) Because HCP also provides the fMRI time course of each component we were able to show that these four networks (1) demonstrate significant covariance with each other and (2) cluster together in a hierarchical arrangement of all 100 components. This is now Figure 3—figure supplement 3.

Figure 3—figure supplement 3 shows that the PD-ICA atrophy map correlates better with 4 ICNs that are inter-connected than with other ICNs. To assess significance we performed permutation testing by randomizing the order of the 100 networks 1000 times and found that the covariance between these first four was significantly greater than chance (p<0.0016).

*4) It is a bit unclear whether the PD-ICA network (*Figure 1*) shows the full ICA component identified combining PD and controls or if only those voxels within the component that showed significant differences between PD and controls. I believe it's the former, but this should be made a bit clearer and it would be helpful to also show that latter. What part of this network shows the greatest difference between PD and controls?*

The PD-ICA map (Figure 1) consists of all voxels in the ICA component (thresholded at z=3) i.e. it is not restricted to the voxels showing a group difference at the voxel level. This is now clarified in the legend. We have generated a second similar figure displaying the T-stat of the group difference at each voxel within the ICA. It is now included as a subfigure of Figure 1 (Figure 1—figure supplement 4).

*5) There are concerns regarding the correlations across the 135 ROIs*. *The authors already have atrophy and connectivity measures at the voxel level so why not do the analysis across voxels?*

We preformed the same correlation as in Figure 4 at a voxel level: i.e. functional connectivity between each voxel and the SN seed region (resting state fMRI data) versus atrophy t-score at each voxel. The correlation coefficient was 0.13, which is highly significant (80,000 voxels). However, we think an anatomical ROI-based connectome is more advantageous. The voxel level connectome suffers from the high spatial correlation of fMRI data between neighbouring voxels, and from the increased noise of voxel as opposed to ROI fMRI data. Also the voxelwise analysis is not possible for a tractography-based connectome.

*By combining different pieces of various brain atlases with their own hand-drawn atlas of brainstem structures, the authors introduce the possibility of bias into their ROI analysis. Is there no suitable existing atlas such as the WFU-Pickatlas? If the authors must use a custom atlas, some criteria regarding which brainstem structures were included versus excluded are needed*.

There is no MRI atlas of the entire brainstem currently available. We agree with the criticism of potential bias. Indeed, when rethinking this issue, we concluded that the atlas regions below the midbrain (caudal to the SN) actually serve no purpose in this study. First, measuring atrophy in the caudal brainstem is difficult using T1-MRI, even with DBM. Second, both rsfMRI and tractography do poorly in this region: fMRI because it is prone to respiratory and pulse artifacts, and tractography because it is difficult to trace white matter pathways to their real target (the brainstem being a dense structure with numerous small nuclei). The brainstem is also especially prone to MRI susceptibility artifacts. These issues are outlined in [32]. Therefore, regions from pons and medulla are expected to contribute mostly noise to the correlation analysis intended to test the epicentre model of disease spread.

We have now repeated all the ROI analyses using only the Hammers and Cerebellum atlases plus three regions not present in these atlases, namely the substantia nigra (SN), subthalamic nucleus (STN) and red nucleus. These three regions were drawn, as before, on the Big Brain (52), and Duvernoy (29) atlases in MNI space. Since our submission we became aware of an MRI atlas of these three structures based on ultra-high resolution Flash MR imaging at 7T (47), which we used to confirm the location and extent of our atlas regions.

This has actually improved our results. Now the SN emerges as a much better disease propagator (see reply to point 8 below). As before, this says nothing about a presumed initial disease epicenter in the medulla (based on Braak). It only suggests that SN is the best propagator of the disease to the supratentorial brain among regions tested.

*6) Atrophy in the PD-ICA, SBR, UPDRS, and age all appear to be somewhat correlated. It would be interesting to know which are independently correlated after accounting for the others using a multivariate analysis*.

*8) The resolution used for the rsfMRI connectivity analyses in healthy young subjects (both seed-based and “propagation model”) is of 3.5mm isotropic, which makes it impossible to distinguish (a seed in) the substantia nigra from the subthalamic nucleus, and probably also the red nucleus. This might explain why the authors found that the latter two structures were as likely to be propagators as the substantia nigra*.

*To alleviate these major concerns, the authors should probably re-do these analyses with an improved resolution dataset, which is for instance readily available in the HCP in a young and healthy population (∼500 subjects at 2mm isotropic)*.

We performed the suggested analysis using the HCP dataset. The results are not different (the three midbrain structures are still equally good propagators), possibly because we are still using the same low-resolution anatomical data for atrophy calculation (i.e. only the fMRI data has the improved resolution). These data are with the new atlas (112 regions, caudal brainstem excluded). Interestingly, we now find a greater correlation with SN in our 51 rsfMRI data. We note a problem with the HCP data: the “FIX’d” resting state data (i.e. corrected for physiological artifacts using FIX) loses much signal in the brainstem and subcortical areas. We therefore chose to retain our original resting state data for this comparison (but we do use HCP to look for resting state ICNs – see above). Here are the correlations (atrophy versus connectivity) for each region and dataset:rsfMRI-51 subjectsrsfMRI-HCPSubstantia Nigra0.40Red Nucleus0.30Subthalamic Nucleus0.28Substantia Nigra0.26Red Nucleus0.28Subthalamic Nucleus0.22Cerebellum Dentate0.27Thalamus0.08Pallidum0.23Cerebellum Vermis VIIb0.07Hippocampus0.22Cerebellum Vermis CrusII0.05Cerebellum Vermis X0.21Cerebellum Fastigial0.05Cerebellum Vermis VIIIa0.20Cerebellum Vermis CrusI0.03Cerebellum Interposed0.20Cerebellum X0.03Cerebellum Fastigial0.20Cerebellum Vermis VI0.02Cerebellum Vermis IX0.20Cerebellum VIIb0.01Cerebellum Vermis VIIIb0.18Cerebellum VI0.01Cerebellum I IV0.18Cerebellum V-0.01Cerebellum Vermis VIIb0.17Cerebellum I IV-0.01Parahippocampal gyrus0.16Cerebellum Interposed-0.01Cerebellum V0.16Cerebellum Vermis VIIIa-0.03Anterior temporal lobe (medial part)0.15Cerebellum VIIIb-0.04Cerebellum Vermis CrusII0.14Cerebellum VIIIa-0.04occipitotemporal gyrus (lateral part)0.14Cerebellum Vermis IX-0.04Cerebellum VIIb0.13Cerebellum Vermis VIIIb-0.05Cerebellum CrusII0.13Cerebellum CrusII-0.06Cerebellum IX0.12Pallidum-0.06Cerebellum VI0.12Cerebellum Dentate-0.08Amygdala0.12Anterior temporal lobe (medial part)-0.09Cerebellum X0.11occipitotemporal gyrus (lateral part)-0.10Cerebellum Vermis CrusI0.10Cerebellum IX-0.10Putamen0.10Occipital lobe (lateral part)-0.12Cerebellum Vermis VI0.08Insula-0.12Cerebellum VIIIa0.07Cuneus-0.12Thalamus0.06Cerebellum Vermis X-0.13Cerebellum VIIIb0.05Cerebellum CrusI-0.14Insula0.04Parahippocampal gyrus-0.14Anterior temporal lobe (lateral part)0.02Caudate nucleus-0.14Cerebellum CrusI0.01Precentral gyrus-0.14Superior temporal gyrus (anterior part)0.01Postcentral gyrus-0.14Caudate nucleus-0.06Putamen-0.16Superior temporal gyrus (posterior part)-0.07Lingual gyrus-0.17Middle and inferior temporal gyrus-0.07Inferior frontal gyrus-0.17Lingual gyrus-0.08Hippocampus-0.18Postcentral gyrus-0.08Parietal lobe (Inferiolateral)-0.18Precentral gyrus-0.09Amygdala-0.19Posterior temporal lobe-0.09Posterior temporal lobe-0.19Inferior frontal gyrus-0.10Superior parietal gyrus-0.21Middle frontal gyrus-0.10Superior temporal gyrus (posterior part)-0.21Cuneus-0.10Middle frontal gyrus-0.22Anterior cingulate gyrus-0.12Anterior temporal lobe (lateral part)-0.23Occipital lobe (lateral part)-0.12Anterior cingulate gyrus-0.23Lateral orbital gyrus-0.16Superior temporal gyrus (anterior part)-0.23Superior frontal gyrus-0.16Posterior cingulate gyrus-0.24Parietal lobe (Inferiolateral)-0.16Lateral orbital gyrus-0.28Superior parietal gyrus-0.20Nucleus accumbens-0.28Pre-subgenual frontal cortex-0.20Middle and inferior temporal gyrus-0.31Posterior orbital gyrus-0.23Straight gyrus-0.33Posterior cingulate gyrus-0.23Subcallosal area-0.34Medial orbital gyrus-0.27Posterior orbital gyrus-0.36Straight gyrus-0.31Medial orbital gyrus-0.36Anterior orbital gyrus-0.33Pre-subgenual frontal cortex-0.36Subgenual frontal cortex-0.34Superior frontal gyrus-0.37Nucleus accumbens-0.38Subgenual frontal cortex-0.41Subcallosal area-0.42Anterior orbital gyrus-0.44

*9) The authors should explain clearly how they manually defined their ROI in the substantia nigra and other small structures (only names of anatomical atlases are specified in the Methods), and extensively discuss the inherent limitations coming with such a resolution for both their seed-based analysis and propagation model*.

In the revised manuscript, we only use three manually delineated structures (SN, STN, Red Nucleus), and explain the method more clearly (please see: “These two atlases do not have adequate segmentation […] and excludes all brainstem regions caudal to the SN”).

*10) Similarly, in the subsection “Spatial Analysis of PD-ICA network”, it is not clear that the location of the T1 weighted results obtained from DBM can be so precisely identified (substantia nigra vs. subthalamic nucleus, PPN, bed nucleus of the stria terminalis, etc), so the authors should make this point clear in their manuscript*.

Regarding the SN, the atrophy map (Figure 1) convincingly shows the highest peak in the SN. In answer to these two points we added the following caveat*:* “Note, however, that spatial resolution limitations for all of the imaging modalities […] may be interpreted as volume changes by the DBM methodology.”

*11) The spatial cross-correlation between the 3 different networks seems to some extent arbitrarily set up at |r|>0.25. The authors should possibly report whether other significant cross-correlations were found for |r|<=0.25. The authors might also want to use the higher dimension ICA decomposition (d=70 instead of d=20) provided in*
[66]*, as their high dimension ICA yields more specific basal ganglia networks.*

Please see our response to concern #3.

12) Regarding the DBM analysis, the use of ICA is quite ingenious, especially considering the multi-centric aspect (16 sites, 3 different scanners) of this imaging cohort. Could the authors please specify whether they found site-specific or scanner-specific artefactual ICs in their results?

To investigate whether there is an effect of center, multivariate analysis was used. The analysis was used for each obtained DBM-network separately using DBM ∼ Group (PD/Control) + Age + Gender + Site. There was no significant effect of site after correcting for multiple comparisons (p > 0.1). This is now included in the Results section.

*What about a direct comparison of their DBM maps between the 2 populations? Presumably, this provided no significant result, which therefore sends a strong methodological message about an optimised approach for multi-centric T1 weighted volumetric studies*.

This is correct. When correcting for multiple comparisons, the univariate voxel-wise approach yields few significant results (shown in Figure 8).

Author response image 4.**DOI:**
http://dx.doi.org/10.7554/eLife.08440.026

*13) There is concern about the specific use of MELODIC on structural data. The reason is that MELODIC is “tuned” to identifying sparse data and is inherently more suited for fMRI rather than structural data. It can therefore be the case that some more “global” components explaining the largest variance across the subjects can be missed sometimes. It seems unlikely here as the inputs used for TICA were DBM maps and not GM maps, and the main IC is reassuringly highly relevant to the pathology studied. It might be worth however for the authors to run FLICA (another data-driven ICA tool available in FSL) on their data to make sure results are similar*.

We confirm that the DBM data posses the super-Gaussian (high kurtosis) quality which is a precondition for using MELODIC. As mentioned above, GM maps and DBM maps have different data structure. GM maps from VBM having a sub-Gaussian distribution in general (due to spatial smoothing), do not possess the sparsity of fMRI data and indeed when we apply MELODIC to VBM maps from the PPMI dataset we obtain a single component (basically the entire brain). DBM data on the other hand are even sparser than fMRI and this makes MELODIC suitable for analyzing them. In order to formally check for sparsity we performed a kurtosis analysis of the PPMI DBM and VBM data as well as our fMRI dataset for comparison. In Figure 9, we present the distributions of kurtosis values computed for each subject.

Author response image 5.**DOI:**
http://dx.doi.org/10.7554/eLife.08440.027

FLICA has recently been used by the FMRIB group to analyze VBM data. We contacted the first author, Dr Douaud, who states that FLICA is more tuned to sub-Gaussian signals, which explains why it succeeds with VBM data. Presumably this would make it less useful for DBM with its high sparsity. We added the following text to the Methods: “The ICA algorithm in MELODIC is sensitive to sparsely distributed (super-Gaussian) data (26) as typically seen in fMRI. The DBM data used here possessed this super-Gaussian property (kurtosis >4).”

14) Regarding their correlations with clinical measures, could the authors maybe justify in the manuscript why they have not used the MoCA and other parts of the UPDRS than part III, or alternatively carry out these correlation analyses?

PPMI has numerous clinical assessment tools. We needed to limit the number of correlations investigated to those measures of disease severity that might best predict brain atrophy to avoid a multiple comparison problem. The two best measures of disease severity are UPDRS part III and SPECT binding (SBR).

The UPDRS has four parts but only part III is an objective assessment of motor function. Parts I and II refer to the patient’s subjective experience of the disease (non-motor and motor). Part I loads onto sleep problems, depression and cognitive impairment. Part II refers to activities of daily living such as speaking, dressing and eating. Example questions are “do you have trouble remembering things or paying attention” or “do you have problems dressing.” While these quality of life assessments are quite relevant to clinical practice, they are likely less closely correlated to neurodegeneration than UPDRS part III, in which a trained examiner measures motor function objectively. Part IV is not relevant as it refers to complications of medications, and all the participants here were unmedicated.

The reason not to study MoCA was different. Entry into the trial consisted mostly of non-demented individuals. The mean MoCA was 27.1 (‘normal’ = 27-30), and only 20% of patients had a score in the ‘abnormal’ range (MoCA<26). There may therefore not be enough variability in MoCA to expect correlations.

Moreover, one paper has already been published on MoCA and cortical thickness in the PPMI database (although only 123 patients were included) (55). Not surprisingly, there was cortical thinning in cognitively impaired patients.

Nonetheless we performed the following correlations:

First we investigated the relationship between the deformation in the PD-ICA network and MOCA score in both patients and controls. There was a correlation between MoCA and atrophy for the two groups together (r=0.13, p=0.01) however this relationship disappeared after accounting for the effect of age (r=-.002, p=0.96). This phenomenon held when examining PD and control subjects separately, with age as a confound (i.e. the correlation between atrophy and MoCA was entirely explained by age in both groups).

We also investigated the relationship between MOCA and deformation in the remaining 29 networks, controlling for age, and again there were no significant correlations, even at a trend p=0.1 level.

In summary, the MoCA predicted brain atrophy, but the effect was entirely age-related, and there were no differences between PD and controls in any of the brain networks. This is likely due to the fact that most patients had a MoCA in the normal range.

*15) Could the authors please provide the list of best propagators using diffusion imaging (similar to*
Table 3*)?*

The list is provided here and in the revised manuscript. With diffusion imaging, SN remains the best propagator except for cerebellar regions. The presence of many cerebellar regions as good propagators in the diffusion-based connectome is possibly artifactual. Cerebellar fibres pass through the pons, forming the cortico-ponto-cerebellar tract. Tractography may be poor at resolving these cortico-ponto-cerebellar connections from ascending fibres whose origin is in brainstem nuclei, and from corticospinal tract fibres, due to partial volume effects (contamination among numerous densely packed adjacent pathways) and decussation within the brainstem. Thus the brainstem and cerebellar portion of the connectome is likely to suffer from imprecise assignment of edges. This could explain the low specificity obtained for the different cerebellar regions as possible propagators (i.e. all cerebellum regions are grouped with very similar correlation values, according to the DW-MRI data). Nonetheless we now present the data and discuss potential pitfalls.rsfMRIDW-MRISubstantia Nigra0.40Cerebellum VIIb-0.28Subthalamic Nucleus0.28Substantia Nigra-0.28Red Nucleus0.28Cerebellum X-0.28Cerebellum Dentate0.27Cerebellum VIIIa-0.27Pallidum0.23Cerebellum VIIIb-0.27Hippocampus0.22Cerebellum Vermis VIIb-0.27Cerebellum Vermis X0.21Cerebellum CrusII-0.26Cerebellum Vermis VIIIa0.20Cerebellum Vermis VIIIa-0.25Cerebellum Interposed0.20Cerebellum Vermis CrusII-0.25Cerebellum Fastigial0.20Cerebellum CrusI-0.25Cerebellum Vermis IX0.20Cerebellum Vermis VIIIb-0.24Cerebellum Vermis VIIIb0.18Cerebellum Vermis CrusI-0.24Cerebellum I IV0.18Cerebellum IX-0.23Cerebellum Vermis VIIb0.17Cerebellum VI-0.23Parahippocampal gyrus0.16Cerebellum Vermis IX-0.23Cerebellum V0.16Cerebellum Vermis VI-0.22Anterior temporal lobe (medial part)0.15Cerebellum Dentate-0.21Cerebellum Vermis CrusII0.14Cerebellum Interposed-0.20occipitotemporal gyrus (lateral part)0.14Cerebellum Fastigial-0.19Cerebellum VIIb0.13Cerebellum V-0.18Cerebellum CrusII0.13Parahippocampal gyrus-0.18Cerebellum IX0.12Cerebellum I IV-0.18Cerebellum VI0.12Cerebellum Vermis X-0.18Amygdala0.12Middle and inferior temporal gyrus-0.15Cerebellum X0.11Occipital lobe (lateral part)-0.14Cerebellum Vermis CrusI0.10Lingual gyrus-0.14Putamen0.10occipitotemporal gyrus (lateral part)-0.13Cerebellum Vermis VI0.08Posterior temporal lobe-0.13Cerebellum VIIIa0.07Amygdala-0.13Thalamus0.06Anterior temporal lobe (medial part)-0.12Cerebellum VIIIb0.05Subthalamic Nucleus-0.11Insula0.04Cuneus-0.10Anterior temporal lobe (lateral part)0.02Red Nucleus-0.09Cerebellum CrusI0.01Hippocampus-0.09Superior temporal gyrus (anterior part)0.01Anterior temporal lobe (lateral part)-0.08Caudate nucleus-0.06Superior temporal gyrus (posterior part)-0.06Superior temporal gyrus (posterior part)-0.07Parietal lobe (Inferiolateral)-0.05Middle and inferior temporal gyrus-0.07Superior temporal gyrus (anterior part)-0.01Lingual gyrus-0.08Thalamus0.02Postcentral gyrus-0.08Superior parietal gyrus0.03Precentral gyrus-0.09Pallidum0.03Posterior temporal lobe-0.09Insula0.12Inferior frontal gyrus-0.10Putamen0.14Middle frontal gyrus-0.10Caudate nucleus0.17Cuneus-0.10Postcentral gyrus0.18Anterior cingulate gyrus-0.12Nucleus accumbens0.20Occipital lobe (lateral part)-0.12Subcallosal area0.23Lateral orbital gyrus-0.16Posterior orbital gyrus0.23Superior frontal gyrus-0.16Posterior cingulate gyrus0.24Parietal lobe (Inferiolateral)-0.16Medial orbital gyrus0.27Superior parietal gyrus-0.20Straight gyrus0.29Pre-subgenual frontal cortex-0.20Subgenual frontal cortex0.29Posterior orbital gyrus-0.23Lateral orbital gyrus0.31Posterior cingulate gyrus-0.23Precentral gyrus0.32Medial orbital gyrus-0.27Inferior frontal gyrus0.32Straight gyrus-0.31Anterior orbital gyrus0.33Anterior orbital gyrus-0.33Superior frontal gyrus0.35Subgenual frontal cortex-0.34Anterior cingulate gyrus0.35Nucleus accumbens-0.38Middle frontal gyrus0.35Subcallosal area-0.42Pre-subgenual frontal cortex0.36

*It would also be interesting to discuss the strong negative correlations reported in*
Table 3.

A significant negative correlation (in the rsfMRI graph) is more difficult to interpret than a positive one (which identifies best propagators of the disease). It could indicate a region affected at later disease stages: e.g. if the disease marches in a single direction, a region at the edge of the advancing wave would exhibit negative correlation. Another possibility is that a region does not accumulate the pathogenic protein, or accumulates but does not propagate: nodes connected to it would be protected. This is speculative however.